

# A generalized construction of Calabi-Yau models and mirror symmetry

**Per Berglund**[1,2][⋆] **and Tristan Hubsch**[3]

**1** Department of Physics, University of New Hampshire, Durham, NH 03824, USA
**2** Theoretical Physics Department, CERN, CH-1211 Geneve, Switzerland
**3** Department of Physics and Astronomy, Howard University, Washington, DC 20059, USA

⋆ per.berglund@unh.edu

## Abstract

We extend the construction of Calabi-Yau manifolds to hypersurfaces in non-Fano toric varieties, requiring the use of certain Laurent defining polynomials, and explore the phases of the corresponding gauged linear sigma models. The associated non-reflexive and non-convex polytopes provide a generalization of Batyrev's original work, allowing us to construct novel pairs of mirror models. We showcase our proposal for this generalization by examining Calabi-Yau hypersurfaces in Hirzebruch n-folds, focusing on n=3,4 sequences, and outline the more general class of so-defined geometries.

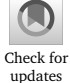
# 1  Incentives, results and summary

A recent study [1] generalized the well-known Calabi-Yau complete intersections of hypersurfaces in products of projective spaces [2–4] so as to allow some of the defining equations to have negative degrees over some of the factor projective spaces, and so necessarily use Laurent (rational) monomials in the defining equations [1,5].[1] Given the novelty of such models and the physics phenomena they exhibit already within the ($m \geqslant 0$) infinite sequence of hypersurfaces in Hirzebruch $n$-folds $\mathscr{F}_m^{(n)}$,[2] in Section 2 we analyze the gauged linear $\sigma$-model (GLSM) world-sheet field-theory [9,10] corresponding to these sequences, focusing on $n = 3, 4$. We also map out the associated enlarged (complete) Kähler moduli space, i.e., "phases," by considering all possible triangulations of the spanning polytope (and its convex hull) associated to the embedding non-Fano toric variety.

In order to analyze the GLSM ground states, which form a toric variety, the secondary fan [9,11–14], in Section 3 we generalize the toric methods [15–20] to cases where Laurent superpotentials naturally appear in the defining equations of the (Calabi-Yau) sub-varieties. In particular, our extension includes a large class of non-convex and possibly self-crossing (VEX) polytopes and corresponding fans, which contain flipped, i.e., reversely oriented cones (faces). We then show that such generalizations: (**a**) produce Laurent monomials in the defining equations of transversal[3] Calabi-Yau hypersurfaces, and (**b**) automatically realize natural pairs of mirror Calabi-Yau $n$-folds generalizing earlier results [21–23], associated to "trans-polar" pairs of VEX polytopes.

In Section 4 we explicitly compute the Euler and Hodge numbers of the Calabi-Yau hypersurfaces in $\mathscr{F}_m^{(n)}$ to demonstrate that these key numerical invariants of the trans-polar pairs of *oriented* VEX polytopes (**a**) evaluate exactly as they do for convex polytopes, and (**b**) exhibit all requisite aspects of the mirror relations. Section 5 summarizes our results and concluding comments, while computational details are collected in the appendices. While this proof-of-concept paper illustrates the various toric geometry techniques by focusing on Hirzebruch $n$-folds [5] and their Calabi-Yau hypersurfaces, more general examples and further details may be found in the companion paper [24].

---

[1] By now, these constructions have a rigorous formulation within the Čech cohomology framework [6].

[2] Using methods of classical algebraic geometry [7,8], we have found that the classical topological data of certain sequences of such constructions exhibit a periodicity [5], which is broken by quantum effects. Since classical physics models on the Calabi-Yau hypersurfaces in the Hirzebruch $n$-fold $\mathscr{F}_m^{(n)}$ are equivalent to those in $\mathscr{F}_{m+n}^{(n)}$, respectively, the transformation $\mathscr{F}_m^{(n)} \to \mathscr{F}_{m+n}^{(n)}$ is a classical (discrete) symmetry. Its breaking by quantum effects such as instanton numbers [5] then represents a novel (stringy) quantum anomaly.

[3] A function $f(x)$ over the toric variety $X$ is *transversal*, i.e., $\Delta_X$-regular [21] if the "base-locus" $\{f(x) = 0 = df(x)\}$ is absent from $X$, such as $x = 0$ from $\mathbb{P}^n = (\mathbb{C}^{n+1} \smallsetminus \{0\})/\mathbb{C}^*$; see Appendix A for more details. The zero-locus $f^{-1}(0) := \{x : f(x) = 0\}$ is then also called transversal. Throughout this paper, we focus on this distinctly algebraic quality, and defer its relation to the subtler complex-analytic property of smoothness (related to Cauchy-Riemann and similar conditions) for later.

## 2 The gauged linear sigma model

Recent work [1, 5] has shown that there are significant merits to constructing Calabi-Yau algebraic varieties at least some of the defining equations of which contain Laurent monomials, and that standard methods of algebraic geometry and cohomological algebra can be adapted to compute the requisite classical data. For applications in string theory and its M- and F-theory extensions, it is desirable to find a world-sheet field theory model with such target spaces.

For well over two decades now, the standard vehicle to this end is Witten's gauged linear sigma model (GLSM) [9, 25, 26], where fermionic integration leaves a potential for the scalar fields of the general form:

$$U(x_i, \sigma_a) = \sum_i \left| F_i \right|^2 + \frac{1}{2e^2} \sum_a D_a{}^2 + \frac{1}{2} \sum_{a,b} \bar{\sigma}_a \, \sigma_b \sum_i Q_i^a Q_i^b |x_i|^2, \tag{1a}$$

$$D_a = -e^2 \Big( \sum_i Q_i^a |x_i|^2 - r_a \Big). \tag{1b}$$

Here $\sigma_a$ is the scalar field from the $a^{\text{th}}$ gauge twisted-chiral superfield, $x_i$ and $F_i$ are respectively the scalar and auxiliary component fields from the $i^{\text{th}}$ "matter" chiral superfield $X_i$, $Q_i^a$ is the charge of the $i^{\text{th}}$ chiral superfield with respect to the $a^{\text{th}}$ $U(1)$ gauge interaction, and the $r_a$ are the contributions from the Fayet-Iliopoulos terms. In supersymmetric theories and especially when acting on chiral superfields, gauge groups are typically complexified and the GLSM naturally has $U(1, \mathbb{C}) \simeq \mathbb{C}^*$ actions — which are the "torus actions" in the toric geometry of the space of ground-states in the GLSM.

### 2.1 Laurent superpotentials

For illustration, consider the GLSM models with the superpotential[4]

$$W(X) := X_0 \cdot f(X), \tag{2a}$$

$$f(X) := \sum_{j=1}^2 \left( \sum_{i=2}^n \big( a_{ij} X_i^n \big) X_{n+j}^{2-m} + a_j X_1^n X_{n+j}^{(n-1)m+2} \right), \tag{2b}$$

where $m, n > 1$ are integers and $X_0$ is the chiral superfield that in some ways serves as a Lagrange multiplier; we focus on $n = 2, 3, 4$, but generalizations are straightforward. Such superpotentials are strictly invariant with respect to the $U_1(1){\times}U_2(1)$ gauge symmetry with the charges

$$Q_1(X_0; X_1, X_2 \cdots, X_n, X_{n+1}, X_{n+2}) = \big( \quad -n; \quad 1, 1, \cdots, 1, 0, 0 \big), \tag{3a}$$

$$Q_2(X_0; X_1, X_2 \cdots, X_n, X_{n+1}, X_{n+2}) = \big( m-2; -m, 0, \cdots, 0, 1, 1 \big). \tag{3b}$$

Manifestly, for $m > 2$, the $a_{ij}$-terms become Laurent monomials; as we will see below in more detail, this turns out to be closely related to the by now very well understood models of Ref. [1, 5, 6], generic examples of which are known to be smooth. For now, we discuss the GLSM with the potential (1)–(2) in its own right, being especially interested in the novelty of the $m > 2$ cases.

The standard requirement for the superpotential to be chiral is straightforwardly satisfied:

$$\bar{D}_{\dot{\alpha}} W(X) = f(X) \big( \underbrace{\bar{D}_{\dot{\alpha}} X_0}_{=0} \big) + \sum_{i=1}^{n+2} X_0 \frac{\partial f(X)}{\partial X_i} \big( \underbrace{\bar{D}_{\dot{\alpha}} X_i}_{=0} \big) = 0, \tag{4}$$

---

[4]This is not the most generic superpotential but the natural generalization of Fermat-like potentials for the current class of models we are considering; see below.

owing to the fact that $X_0$ and all $X_i$'s are chiral superfields, and regardless of the fact that the chiral superfields $X_{n+1}$ and $X_{n+2}$ appear with negative powers for $m > 2$: As we will show below, the background values (vev's) of the lowest component fields in $X_0, X_i$ are always restricted so that the vev's of $f(X)$ and $X_0 \frac{\partial f(X)}{\partial X_i}$ remain finite, even in the cases when $\langle X_{n+j} \rangle \to 0$; see Appendix A. Eq. (4) then insures that the superpotential (2) is itself a chiral superfield and all manifest supersymmetry methods apply; we therefore proceed as usual. Expanding about these vev's makes this superpotential regular in all component fields, and so insures that (2) specifies at least the low-energy regime of these models, with a search for a suitable UV completion beyond our present scope.[5]

## 2.2 The ground state

The potential (1) is a sum of positive-definite terms, each of which has to vanish separately in the ground state. The first four groups of constraints stem from the vanishing of the $F$-terms, for which the equations of motion give $F_i = \frac{\partial W}{\partial x_i}$:

$$\left| \frac{\partial W(x)}{\partial x_0} \right|^2 = \left| \underbrace{\sum_{j=1}^{2} \left( \left( \sum_{i=2}^{n} \frac{a_{ij} x_i^n}{x_{n+j}^{m-2}} \right) + a_j x_1^n x_{n+j}^{(n-1)m+2} \right)}_{= f(x)} \right|^2 \overset{!}{=} 0. \tag{5a}$$

$$\left| \frac{\partial W(x)}{\partial x_1} \right|^2 = |x_0|^2 \left| \underbrace{n x_1^{n-1} \left( \sum_{j=1}^{2} a_j x_{n+j}^{(n-1)m+2} \right)}_{= \partial f(x)/\partial x_1} \right|^2 \overset{!}{=} 0, \tag{5b}$$

$$\left| \frac{\partial W(x)}{\partial x_i} \right|^2 = |x_0|^2 \left| \underbrace{n x_i^{n-1} \left( \sum_{j=1}^{2} \frac{a_{ij}}{x_{n+j}^{m-2}} \right)}_{= \partial f(x)/\partial x_i} \right|^2 \overset{!}{=} 0, \quad i = 2, \ldots, n; \tag{5c}$$

$$\left| \frac{\partial W(x)}{\partial x_{n+j}} \right|^2 = |x_0|^2 \left| \underbrace{(2-m) \sum_{i=2}^{n} \frac{a_{ij} x_i^n}{x_{n+j}^{m-1}} + \left( (n-1)m+2 \right) a_j x_1^n x_{n+j}^{(n-1)m+1}}_{= \partial f(x)/\partial x_{n+j}} \right|^2 \overset{!}{=} 0, \quad j = 1, 2; \tag{5d}$$

For $m \geqslant 3$, the defining constraints (5a), (5c) and (5d) include rational monomials, which are discussed in Appendix A. The vanishing of the last term in (1) imposes:

$$\left| (-n)\sigma_1 + (m-2)\sigma_2 \right|^2 |x_0|^2 + \left| \sigma_1 + (-m)\sigma_2 \right|^2 |x_1|^2 + |\sigma_1|^2 \sum_{i=2}^{n} |x_i|^2 + |\sigma_2|^2 \sum_{j=1}^{2} |x_{n+j}|^2 \overset{!}{=} 0. \tag{5e}$$

This identifies the "normal mode" linear combinations: $U_3(1)$ generated by $mQ_1 + Q_2$ with respect to which $x_1$ is neutral, and $U_4(1)$ generated by $(m-2)Q_1 + nQ_2$ with respect to which $x_0$ is neutral.

Finally, the vanishing of the $D$-terms (1) impose:

$$\frac{e^2}{2} \left( -n|x_0|^2 + \sum_{i=1}^{n} |x_i|^2 - r_1 \right) \overset{!}{=} 0, \tag{6a}$$

$$\frac{e^2}{2} \left( (m-2)|x_0|^2 - m|x_1|^2 + \sum_{j=1}^{2} |x_{n+j}|^2 - r_2 \right) \overset{!}{=} 0. \tag{6b}$$

---

[5]While completing this article, we have learned from Lara Anderson and James Gray of their as yet unpublished exploration of similar superpotentials, in the context of adapting the GLSM to the gCICYs [1] as well as of recent work on $(0, 2)$ GLSMs [27]; it is our understanding that these analyses are mutually consistent, complementary and corroborating.

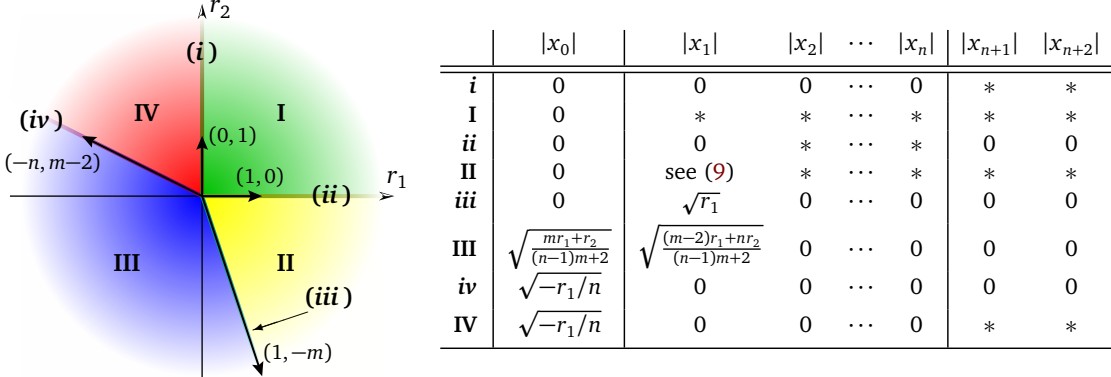

Figure 1: The phase diagram of the GLSM with the Calabi-Yau $n$-fold $\subset \mathscr{F}_m^{(n)}$ "geometric" phase; the "$*$" entries are generally nonzero and are outside the Stanley-Reisner ideal.

## 2.3 Phases

We now turn to analyze the D-term constraints (6), following [14]. The $U(1)$ charges $Q_i^a$ of the chiral superfields, $(X_0, X_i)$, determine the two-dimensional secondary fan (phase diagram) given in Figure 1.[6] In particular, we can find the phase-boundaries by determining the conditions for the $U(1)^2 \to U(1)$ gauge symmetry breaking. When[7] (*i*): $x_0 = 0 = x_i$ for $i = 1, \ldots, n$ but $x_{n+j} \neq 0$ for $j = 1, 2$, the $U_1(1)$ gauge group is preserved but $U_2(1)$ is broken completely. In this case (6a) and (6b) imply that $r_1 = 0$ and $r_2 \geq 0$, respectively; this happens along the $(0, 1)$-direction in the $(r_1, r_2)$-plane. Similarly, (*ii*): $U_2(1)$ is preserved when $x_0 = 0 = x_1 = x_{n+1} = x_{n+2}$ but $x_i \neq 0$ for $i = 2, \cdots, n$, while $U_1(1)$ is broken completely. Then, $r_2 = 0$ and $r_1 \geq 0$ from (6b) and (6a), respectively; this happens along the $(1, 0)$-direction. Next, (*iii*): if only $x_1 \neq 0$, the combined $U_3(1)$ gauge symmetry generated by the charges $mQ_1 + Q_2$ is preserved, so (6a) constrains $r_1 \geq 0$ and the D-term from this $U_3(1)$ implies that $m\, r_1 + r_2 = 0$; this happens along the $(1, -m)$-direction. Finally, (*iv*): if only $x_0 \neq 0$, the combination $U_4(1)$ gauge symmetry generated by the charges $(m-2)Q_1 + nQ_2$ is preserved. The corresponding D-term constraint, in terms of the corresponding combination of (6a) and (6b), implies that $(m-2)r_1 + n\, r_2 = 0$. Also, (6a) then implies that $r_1 \leq 0$; this happens along the $\left(-n, (m-2)\right)$-direction.

Thus, there are four different phases, as depicted in Figure 1. We now analyze them in turn, using that a ground state solution must also satisfy the $F$-term constraints (5), as detailed in Appendix A.

**Phase I:** $r_1, r_2 > 0$. The $F$-term constraints are solved by having $x_0 = 0$ and $f(x) = 0$. From the D-term analysis above, the *excluded* region in the field-space

$$\mathscr{I}_I = \{x_1 = \ldots = x_n = 0\} \cup \{x_{n+1} = x_{n+2} = 0\} \tag{7}$$

is exactly the Stanley-Reisner (or irrelevant [19]) ideal for the Hirzebruch $n$-fold $\mathscr{F}_m^{(n)}$ ($m$-twisted $\mathbb{P}^{n-1}$-bundle over $\mathbb{P}^1$). Since the $x_{n+j}$ cannot both vanish (5e) implies that $\sigma_2 = 0$. Eq. (5e) then simplifies and implies that $\sigma_1 = 0$ since the $x_i$, $i = 1, \ldots, n$ cannot all be zero. Thus, $f(x) = 0$ defines a Calabi-Yau $(n-1)$-fold hypersurface in $\mathscr{F}_m^{(n)}$.

---

[6]The analysis of the secondary fan, referred to as the enlarged Kähler moduli space can also be done by first considering the toric variety, $X$, which is the ambient space for the Calabi-Yau hypersurface, for more details, see section 3. There are four different triangulations of the associated spanning polytope $\Delta_X^\star$, not necessarily containing all of the points in $\Delta_X^\star$, which allows us to to obtain the large radius Calabi-Yau phase as well as the Landau-Ginzburg orbifold [13].

[7]For the remainder of this section, we omit writing the vacuum expectation bra-kets for brevity, so "$x_i = 0$" will denote the vanishing of the vev $\langle x_i \rangle$, not the field.

Direct computation shows that the polynomial $f(x)$ is transversal for generic choices of $a_{ij}, a_j$, so that its $n+2$ gradient components $\frac{\partial f}{\partial x_i}, \frac{\partial f}{\partial x_{n+j}}$ vanish simultaneously with $f(x)$ itself only within the excluded region (7), see Appendix A for more details.

**Phase II:** $-mr_1 < r_2 < 0$. The $F$-term constraints are still solved by having $x_0 = 0$ and $f(x) = 0$. From the $D$-term analysis above, the *excluded* region in the field-space

$$\mathcal{I}_{II} = \{x_1 = 0\} \cup \{x_2 = \ldots = x_{n+2} = 0\} \tag{8}$$

is the Stanley-Reisner ideal for the weighted projective space $\mathbb{P}^n_{(m:\cdots:m:1:1)}$ in terms of the co-ordinates $(x_2, \ldots, x_{n+2})$. With $x_1 \neq 0$, (5e) implies that $\sigma_1 = m\sigma_2$, and since the remaining $x_i$ cannot all vanish simultaneously, it follows that $\sigma_1 = \sigma_2 = 0$. Thus, $f(x) = 0$ defines (the MPCP-desingularization of) the Calabi-Yau $(n-1)$-fold hypersurface $\mathbb{P}^n_{(m:\cdots:m:1:1)}[(n-1)m+2]$. Indeed, Eqs. (6a) and (6b) imply that (recall that $r_2 < 0$)

$$|x_1| = \sqrt{\frac{\sum_j |x_{n+j}|^2 - r_2}{m}} = \sqrt{r_1 - \sum_{i=2}^{n} |x_i|^2} > 0 \tag{9}$$

throughout this phase, and $x_1$ parametrizes the exceptional set of the MPCP-desingularization of $\mathbb{P}^n_{(m:\cdots:m:1:1)}$. Clearly, $|x_1| \to 0$ at the boundary (*ii*) where $x_{n+1}, x_{n+2}$ and $r_2$ vanish, while $|x_1| \to \sqrt{r_1} > 0$ on the boundary (*iii*) where $x_i = 0$ but $r_1 > 0$. It then follows that at the boundary (*ii*) the MPCP-desingularization is "blown-down," leaving the hypersurface $\mathbb{P}^n_{(m:\cdots:m:1:1)}[(n-1)m+2]$ with the unresolved orbifold singularity — which gives rise to the partially restored $U_2(1)$ gauge symmetry indicated above.

Within the phase II region and away from its boundaries, both $U(1)$-symmetries are completely broken over *most* of the $f(x) = 0$ surface when $x_1, \cdots, x_n \neq 0$. However, at special points where some but not all of $x_1, \cdots, x_n$ vanish, a discrete subgroup of the gauge symmetry may be restored, such as at $(x_1, 0, \cdots, 0, x_{n+1}, \omega x_{n+1})$ with $\omega^{(n-1)m+2} = -\frac{a_1}{a_2}$ to satisfy (5)–(6); here $x_{n+1} \neq 0$ completely breaks $U_2(1)$, whereupon $x_1 \neq 0$ breaks $U_1(1) \to \mathbb{Z}_n$; these are then "local" $\mathbb{Z}_n$ orbifold points of the hypersurface $f(x) = 0$ — which however is not a global $\mathbb{Z}_n$-quotient. Phase II is thus identified as the "orbifold phase" [13, 14].

**Phase III:** $\frac{m-2}{n} r_1 < r_2 < -mr_1$. The D-term constraints imply that now both $x_0 = 0$ and $x_1 = 0$ must be *excluded*:

$$\mathcal{I}_{III} = \{x_0 = 0\} \cup \{x_1 = 0\}. \tag{10}$$

Thus, all the $F$-term constraints, (5b)-(5a) are solved by setting $x_i = 0$ for $i = 2, \cdots, n+2$, and (5e) simplifies to

$$\left|(-n)\sigma_1 + (m-2)\sigma_2\right|^2 |x_0|^2 + \left|\sigma_1 + (-m)\sigma_2\right|^2 |x_1|^2 \overset{!}{=} 0. \tag{11}$$

The vevs $x_0 \neq 0 \neq x_1$ break $U_1(1) \times U_2(1) \to \mathbb{Z}_{m(n-1)+2}$; e.g., $x_1 \neq 0$ sets $\sigma_1 = m\sigma_2$, producing $\left|(m(1-n)-2)\sigma_1\right|^2 |x_0|^2 = 0$. Thus, the vacuum solution in phase III is that of a $\mathbb{Z}_{m(n-1)+2}$ Landau-Ginzburg orbifold of $f(x) = 0$, acting with charges $(mQ_1 + Q_2)$.

**Phase IV:** $0 < r_2 < \frac{m-2}{n} r_1$. The $D$-term analysis implies that $x_0 = 0$ must now be *excluded*, whereupon the first two $F$-term constraints, (5b) and (5c), imply that $x_1 = \ldots = x_n = 0$. The remaining $F$-term constraints, (5a) and (5d) turn out to be satisfied, leaving $x_{n+1}, x_{n+2}$ unconstrained. Since $x_1 = 0$, the 2nd $D$-term constraint (6b) produces $\sum_{j=1}^{2} |x_{n+j}|^2 = r_2 + \frac{m-2}{n} r_1$, which is positive in this phase, so that $x_{n+1} = x_{n+2} = 0$ must also be *excluded*, and we obtain:

$$\mathcal{I}_{IV} = \{x_0 = 0\} \cup \{x_{n+1} = 0 = x_{n+2}\}. \tag{12}$$

| | $x_0$ | Fiber $x_1$ | $x_2$ $\cdots$ $x_n$ | Base $x_{n+1}$ $\quad x_{n+2}$ | Gauge Group | Phase Region |
|---|---|---|---|---|---|---|
| $Q_1$ | $-n$ | $1$ | $1$ $\cdots$ $1$ | $0 \qquad 0$ | $U_1(1)$ | $(r_1,r_2)$-plane |
| $Q_2$ | $m{-}2$ | $-m$ | $0$ $\cdots$ $0$ | $1 \qquad 1$ | $\qquad U_2(1)$ | |
| $i$ | $0$ | $0$ | $0$ $\cdots$ $0$ | $* \qquad *$ | $U_1(1)$ $\qquad$ — | $r_1=0,\ \ r_2>0$ |
| I | $0$ | $\notin\{\{x_i=0\}\cup\{x_{n+j}=0\}\}$ | | | — $\qquad$ — | $r_1>0,\ \ r_2>0$ |
| $ii$ | $0$ | $0$ | $\notin\{x_i=0\}$ | $0 \qquad 0$ | — $\qquad U_2(1)$ | $r_1>0,\ \ r_2=0$ |
| II | $0$ | see (9) | $\notin\{\{x_i=0\}\cup\{x_{n+j}=0\}\}$ | | —$^\star$ $\quad$ —$^\star$ | $r_1>0>r_2>-mr_1$ |
| $iii$ | $0$ | $\sqrt{-\frac{r_2}{m}}=\sqrt{r_1}$ | $0$ $\cdots$ $0$ | $0 \qquad 0$ | $U_3^\dagger(1)\times\mathbb{Z}_M^\ddagger$ | $r_1>0>r_2=-mr_1$ |
| III | $\sqrt{\frac{mr_1+r_2}{(n-1)m+2}}$ | $\sqrt{\frac{(m-2)r_1+nr_2}{(n-1)m+2}}$ | $0$ $\cdots$ $0$ | $0 \qquad 0$ | $\mathbb{Z}_M^\ddagger$ | $-mr_1>r_2>\frac{m-2}{n}r_1$ |
| $iv$ | $\sqrt{-\frac{r_1}{n}}$ | $0$ | $0$ $\cdots$ $0$ | $0 \qquad 0$ | $\mathbb{Z}_M^\ddagger\times U_4^\dagger(1)$ | $r_2=\frac{m-2}{n}r_1>0$ |
| IV | $\sqrt{-\frac{r_1}{n}}$ | $0$ | $0$ $\cdots$ $0$ | $* \qquad *$ | $\mathbb{Z}_n$ | $\frac{m-2}{n}r_1>r_2>0$ |

$^\star$ Generically, the $U(1)^2$ is completely broken, but is "restored" to a discrete subgroup at special points.

$^\dagger$ The combination $U_3(1)$ is generated by $mQ_1+Q_2$, whereas $U_4(1)$ is generated by $(m-2)Q_1+nQ_2$

$^\ddagger$ $M=(n-1)m+2$: in $iii$, $x_1\neq0$ breaks $U_4(1)\to\mathbb{Z}_{(n-1)m+2}$; in $iv$, $x_0\neq0$ breaks $U_3(1)\to\mathbb{Z}_{(n-1)m+2}$

Table 1: The cyclic listing of the phases (I–IV) and boundaries ($i$–$iv$) of the GLSM (2)

Since $x_{n+1}, x_{n+2}$ cannot both vanish, their vevs break $U_2(1)$ completely, while $x_0\neq0$ breaks $U_1(1)\to\mathbb{Z}_n$ since $Q_1(x_0)=-n$; correspondingly, (5e) reduces to $|(-n)\sigma_1||x_0|=0$. This then is a hybrid phase in which a Landau-Ginzburg $\mathbb{Z}_n$ orbifold of $\{(x_1,\cdots,x_n):f(x)=0\}$ is fibered over the base-$\mathbb{P}^1=\mathbb{P}(x_{n+1},x_{n+2})$.

At this point let us make the following comment on the vanishing of the $x_i=0$, $i>0$ in the Landau-Ginzburg phase above and the existence of a superpotential with Laurent monomials (2). By going through phase IV, where $x_i=0$ for $i=1,\ldots,n$ correspond to the fiber having collapsed to a Landau-Ginzburg orbifold, the Laurent monomials are absent — in fact, $f(x)$ vanishes identically in phase IV. By transitioning to phase III, we then have $x_1\neq0$ while $x_2=\cdots=x_{n+2}=0$, which is the true Landau-Ginzburg orbifold. On the other hand, when transitioning from phase II, through the boundary ($iii$), into phase III, we have to specify the intrinsic limit $x_i, x_{n+j}\to0$ so that $f(x)=0$ remains well-defined; see Appendix A. To this end,

$$\lim_{x_1\to0}f(x)=\sum_{i=2}^{n}\Big(\frac{a_{i1}}{x_{n+1}^{m-2}}+\frac{a_{i2}}{x_{n+2}^{m-2}}\Big)x_i^n \tag{13}$$

shows that $x_2,\cdots,x_n$ should vanish sufficiently faster than $x_{n+j}$, except perhaps one of the $x_i$ for which then we must insure that $x_{n+2}=\omega\,x_{n+1}\to0$ with $\omega^{m-2}=-\frac{a_{i2}}{a_{i1}}$.

The phases and their boundaries are summarized in Table 1.

The vevs of $x_0$ and $x_1$ change continuously as $(r_1,r_2)$ are varied through the cycle of phases I–II–III–IV–I, so that the secondary fan depicted to the left in Figure 1 is complete as given.

## 3   The toric geometry of the ground state

We now turn to examine the toric geometry of GLSM ground states defined by Laurent superpotentials such as (2), and exhibit a justification for the inclusion of Laurent monomials in superpotential such as (1) and its generalizations. In particular, Laurent superpotentials such as (2) motivate a refinement of the standard methods of toric geometry [15–20], and we first

sketch a few basic facts to establish notation and conventions, adapting from [17–19].

Every compact toric variety may be specified by a complete rational polyhedral fan $\Sigma$ within a lattice $N$, which in turn has a lattice *spanning polytope*[8] $\Delta^\star$, the $N$-integral points $v_\rho$ of which being the minimal generators of $\Sigma$, and $\Delta^\star$ is (coarsely) star triangulated by $\Sigma$ about $(0 \in \Sigma) \in \mathrm{rel\,int}\,\Delta^\star$. Every toric variety also has a Newton polytope $\Delta$ (in a lattice $M$ dual to $N$), the $M$-integral points of which correspond to anticanonical sections, specified as monomials in the so-called Cox variables [28] $x_\rho \xleftrightarrow{1\text{-}1} v_\rho \in \Delta^\star$:

$$\bigoplus_{\mu \in \Delta \cap M} \Big( \prod_{v_\rho \in \Delta^\star \cap N} x_\rho^{\langle v_\rho, \mu \rangle + 1} \Big) \mapsto H^0(\mathcal{K}^*), \qquad v_\rho \tag{14}$$

where $\langle\,,\,\rangle$ denotes the Euclidean scalar product, and the "+1" in the exponent indicates sections of the anticanonical $(\mathcal{K}^*)^{+1}$. For convex and reflexive Newton polytopes, the spanning polytope $\Delta^\star$ turns out to be the so-called *polar* of the Newton polytope $\Delta$:

$$\Delta^\circ := \big\{ u : \langle v, u \rangle \geqslant -1,\ v \in \Delta \big\}, \tag{15}$$

and the polar operation is involutive: $\Delta = (\Delta^\star = \Delta^\circ)^\circ$. The global nature of the definition (15) also implies that

$$(\Delta^\star)^\circ = \big( \mathrm{Conv}(\Delta^\star) \big)^\circ \quad \text{so} \quad \big( (\Delta^\star)^\circ \big)^\circ = \mathrm{Conv}(\Delta^\star), \tag{16}$$

where $\mathrm{Conv}(\Delta^\star)$ is the *convex hull* (envelope) of $\Delta^\star$. For convex polytopes, $\mathrm{Conv}(\Delta^\star) = \Delta^\star$, but this is not so for *non-convex* polytopes: the global nature of (15) obscures every non-convex detail in non-convex polyhedra such as depicted in Figure 2. Also, for every reflexive convex polyhedron $\Delta^\star$, the $x_\rho$-monomials given by $(\Delta^\star)^\circ$ always turn out to be regular, and so cannot provide for the Laurent monomials appearing in (2b) for $m > 2$.

This "non-convexity hiding" nature (16) of the standard polar operation (15) turns out to be closely correlated with the systematic omission of Laurent $x_\rho$-monomials in (14), which is inadequate for constructing Calabi-Yau hypersurfaces in non-Fano $n$-folds such as considered recently [1, 5, 6]. We thus seek a *twin generalization* of both the polar operation (15) and of (convex) reflexive polytopes.

## 3.1 The generalization

We propose a *twin definition* of a class of "VEX polytopes" (to include all convex reflexive polytopes but also certain non-convex ones,[9]) and a "trans-polar" operation amongst them such that:

    A. For every convex polytope $\Delta$, the trans-polar equals the polar: $\Delta^\triangledown = \Delta^\circ$.
    B. The trans-polar of every VEX polytope is also a VEX polytope.
    C. For every VEX polytope $\Delta$, $(\Delta^\triangledown)^\triangledown = \Delta$.

To avoid confusion, $\Delta^\triangledown$ will denote the trans-polar of $\Delta$, while $\Delta^\circ$ continues to denote its standard polar (15) [15–20]. Extending (15), we propose to define the trans-polar operation by the following (iterative-recursive) face-by-face procedure:

---

    [8]We use Definition V.4.3 and rely on the subsequent Theorem V.4.5 [18, p. 159]. We denote the spanning polytope by $\Delta^\star$, the "$\star$" reminding that the polytope has a star triangulation defined by the fan that it spans.

    [9]The generalization includes novel star triangulations of reflexive polytopes, in which non-star simplices are excluded from the triangulation; see below.

**Construction 3.1** (trans-polar). *Given an integral polytope $\Delta$, with an integral star-triangulation consisting of only unit-degree, $d(\theta) = 1$, star-simplices, $\theta$:*.[10]

1. *Recursively decompose $\Delta$ into a disjoint union of convex faces $\theta_\alpha$, obtained by subdividing any non-convex face in the process; in practice, a subset of this hierarchy suffices.*

2. *Construct the polar $\theta_\alpha^\circ$ to each face $\theta_\alpha \subset \Delta$ using the boundary version of (15), with "=" in place of "⩾";*

3. *Dually to the hierarchy $\{\theta_\alpha\} \subset \Delta$, assemble the $\{\theta_\alpha^\circ\}$ into $\Delta^\nabla$, the trans-polar of $\Delta$, using the dual/polar relations:*

$$
\begin{aligned}
\vartheta = \theta_1 \cap \cdots \cap \theta_k &\quad \Rightarrow \quad \vartheta^\circ = [\theta_1^\circ, \cdots, \theta_k^\circ], \\
\vartheta = [\theta_1, \cdots, \theta_k] &\quad \Rightarrow \quad \vartheta^\circ = \theta_1^\circ \cap \cdots \cap \theta_k^\circ,
\end{aligned} \tag{17}
$$

*where $[\theta_1, \cdots, \theta_N]$ is the face delimited by $\theta_1, \cdots, \theta_N$.*

The degree $d(\theta)$ of a face $\theta$ effectively counts the number of unit-degree star-simplices over $\theta$, see also Appendix B for more details. Each subdivision needed in *Step 1* introduces not-uniqueness, but consists of co-planar convex sub-faces; their polars in *Step 2* coincide, rendering the different subdivisions equivalent. Finally, we use that through *Step 3* of Construction 3.1 (see Section 3.2), a choice of an orientation of $\Delta$ induces an orientation of its trans-polar, $\Delta^\nabla$; this is consistent with the "winding number" of multi-fans [29–32], and provides additional information, such as (24) and (34), that corroborates Construction 3.1.

While we have no formal proof that the specifications in Construction 3.1 always suffice, this does turn out to be true in several dozens of 2- and 3-dimensional examples (including several infinite sequences) purposefully constructed to test it; Section 3.2 presents an illustrative sequence, many more to be presented in Ref. [24]. Suffice it here to note that each of the several dozens of example trans-polar polytopes produced in testing Construction 3.1 span the (**1**) (multi-)fan of internal normal cones [19, p. 76], i.e., (**2**) the reflection through 0 of the fan of external normal cones [18, §I.4], and (**3**) coincides with a complementary construction based on the matrix-representation of star-simplices [8, p. 115–117]. For convex reflexive polytopes, these are indeed alternative constructions of the polar (15) polytope, but turn out to also apply to non-convex VEX polytopes—where they coincide with the results of Construction 3.1 to the extent of our testing; for more details and examples, see Ref. [24].

Requirement A is satisfied by design for all convex polytopes: *Step 1* may stop with $\Delta$ itself since it is convex, *Step 2* produces $\Delta^\nabla = \Delta^\circ$, and there is nothing left for *Step 3*. Relaxing convexity, requirements B and C define the class of VEX polytopes as the maximal closure under the trans-polar operation. By defining the **deficit** of (15):

$$
\mathrm{dfc}(\circ, \Delta) := \left((\Delta)^\circ\right)^\circ \smallsetminus \Delta, \tag{18}
$$

requirement C above is equivalent to requiring the trans-polar operation of Construction 3.1 to have no deficit on VEX polytopes. For any toric variety $X$ and its spanning polytope $\Delta_X^\star$, we define the **extension** part of the (complete) Newton polytope:[11]

$$
\mathrm{xtn}(\Delta_X) := \big(\underbrace{\Delta_X := (\Delta_X^\star)^\nabla}_{\text{complete}}\big) \smallsetminus \big(\underbrace{(\Delta_X^\star)^\circ = \big(\mathrm{Conv}(\Delta_X^\star)\big)^\circ}_{\text{(in)complete}}\big). \tag{19}
$$

---

[10]The degree of a $k$-face is the $k!$-multiple of the $k$-volume, $d(\theta^{(k)}) := k! \cdot \mathrm{Vol}_k(\theta^{(k)})$ [21] Faces $\theta$ with $d(\theta) > 1$ correspond to singular regions in the toric variety, and Construction 3.1 can be extended to include a suitable desingularization; for more detail, see Ref. [24]. Also, more general fan-like structures called "multi-fans" that include overlapping cones have been discussed in Refs. [29–32]. Herein we focus on $n$-dimensional fans which, while possibly flip-folded as in Figure 4, are oriented and effectively cover $\mathbb{R}^n$ exactly once, so we omit the "multi-" prefix; for generalizations, see [24].

[11]We show below that $\mathrm{xtn}(\Delta_X)$ encodes the Laurent monomials for non-convex VEX polytopes $\Delta_X$.
| $\Delta^\star_{\mathscr{F}_m}$ | $v_0$ | $v_1$ | $v_2$ | $v_3$ | $v_4$ | $v_5$ |
|---|---|---|---|---|---|---|
| *fiber* | 0 | −1 | 1 | 0 | 0 | −m |
|  | 0 | −1 | 0 | 1 | 0 | −m |
| *base* | 0 | 0 | 0 | 0 | 1 | −1 |
| $\mathscr{M}(\Delta^\star_{\mathscr{F}_m})\begin{cases} \ell_1 \\ \ell_2 \end{cases}$ | −3 | 1 | 1 | 1 | 0 | 0 |
|  | m−2 | −m | 0 | 0 | 1 | 1 |
|  | $x_0$ | $x_1$ | $x_2$ | $x_3$ | $x_4$ | $x_5$ |

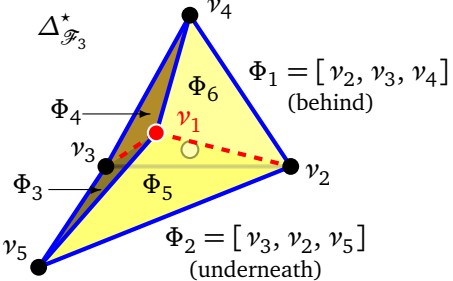

Figure 2: The spanning polytope of the Hirzebruch 3-fold $\mathscr{F}_m$: specified by a chart of its vertices, $v_\rho$, and Mori vectors, $\ell_a$, (left), and depicted for $m = 3$ (right). Here $(x_1, x_2, x_3)$ and $(x_4, x_5)$ are the homogeneous coordinates of the fiber $\mathbb{P}^2$ and base $\mathbb{P}^1$, respectively.

We now proceed to give a more explicit form of the conditions on the VEX polytopes and the construction of mirror manifolds in terms of the associated toric varieties.

▶ Although we have no conclusive explicit definition of VEX polytopes, they are necessarily defined within a lattice $L \simeq \mathbb{Z}^n$ and must have:

1. a single $L$-integral point, 0, in the interior of the polytope $\Delta \subset (L \otimes \mathbb{R})$;
2. only $L$-integral vertices, each at minimal Euclidean distance from 0;
3. no $k$-face ($k > 0$) in a $k$-plane containing 0;
4. a star-triangulation in which $L$-integral points are only at the apex 0 and the base of each star-simplex;
5. the interval $[p, 0] \subset \Delta$ for each $p \in \Delta$, i.e., each VEX polytope is a star-domain in $\mathbb{R}^n$, understood so as to allow for flip-folded facets such as shown in Figure 4, and even multi-fans [29–32] more generally; see Ref. [24] for more examples.

These requirements may be redundant: for 2-dimensional polytopes condition 2 implies condition 3, but not so in higher dimensions; for non-convex polytopes, condition 1 does not imply condition 4; etc.

▶ This generalizes mirror symmetry as applied to Calabi-Yau hypersurfaces constructed from reflexive pairs $(\Delta^\star_X, \Delta_X)$, in which the mirror manifold of a generic Calabi-Yau hypersurface $\hat{Z}_f \subset X$, specified by $f(x) = 0$ in a toric variety $X$ with a reflexive (and convex) Newton polytope $\Delta_X$ is an MPCP desingularization of a suitable finite quotient of the Calabi-Yau hypersurface $\hat{Z}_g \subset X$, specified by $g(y) = 0$ in the toric variety $Y$ the Newton polytope of which is $\Delta_Y = \Delta^\circ_X$ [21, 33]. We hereby extend this to define a class of VEX polytopes wherein every pair of trans-polar polytopes defines a pair of trans-polar toric varieties,

$$(X, Y) : (\Delta^\star_X, \Delta_X) = \left( (\Delta_X)^\nabla, (\Delta^\star_X)^\nabla \right) = \left( (\Delta^\star_Y)^\nabla, (\Delta_Y)^\nabla \right) = (\Delta_Y, \Delta^\star_Y) \tag{20}$$

so that an MPCP desingularization of a suitable finite quotient of the Calabi-Yau hypersurface $\hat{Z}_g \subset Y$ is a natural mirror of a Calabi-Yau hypersurface $\hat{Z}_f \subset X$. The point of this proof-of-concept note is to show that this includes a large collection of non-convex polytopes such as those of Hirzebruch $n$-folds.

## 3.2 A 3-dimensional example: elliptically fibered K3 manifolds

As an illustration, consider the Hirzebruch 3-folds ($m$-twisted $\mathbb{P}^2$-bundles over $\mathbb{P}^1$), specified in Figure 2. The Mori vectors $\ell_a$ give the $U(1)^2$ charges $Q^i_a$ of the GLSM, the analysis of which gives a geometric phase (I), in terms of a K3 hypersurface $f(x) = 0 \subset \mathscr{F}_m$. Furthermore, there is the orbifold phase (II) where the combination of Mori vectors $\ell = m\ell_1 + \ell_2 = (0, m, m, 1, 1)$

specifies the weights of the weighted projective space $\mathbb{P}^3_{(m:m:1:1)}$ to which $\mathscr{F}_m$ may be blown down by eliminating the vertex $\nu_1$, augmenting $\Delta^\star_{\mathscr{F}_m} \to \mathrm{Conv}(\Delta^\star_{\mathscr{F}_m}) = \Delta^\star_{\mathbb{P}^3_{(m:m:1:1)}}$ and formally setting $x_1 \to 1$, which is akin to restricting to a (partial) chart where $x_1 \neq 0$. These weights precisely correspond to the charges (3b) for $n = 3$, in which case (5a) defines a $K3$ surface as the Calabi-Yau hypersurface $\mathbb{P}^3_{(m:m:1:1)}[2(m{+}1)]$. The quasi-projective space $\mathbb{P}^3_{(m:m:1:1)}$ is singular at $(1,1,0,0)$, and the vector $\nu_1$ corresponds to its MPCP desingularization [21].

**The Newton polytope:**    As seen clearly in Figure 2, the polytope with vertices $\{\nu_1, \cdots, \nu_5\}$ fails to be convex at $\nu_1$ for $m > 2$. The standard definition of its polar, owing to (16), produces the Newton polytope of $\mathbb{P}^3_{(m:m:1:1)}$ rather than that of $\mathscr{F}_m$. Furthermore, the Newton polytope of $\mathbb{P}^3_{(m:m:1:1)}$ is depicted by the yellow (shaded) portion of the right-hand side diagram in Figure 3 and has two fractional vertices, $\mathring{\mu}_1 = (\frac{5}{3}, -1, -1)$ and $\mathring{\mu}_2 = (-1, \frac{5}{3}, -1)$.[12] Its $M$-integral points correspond through (14) to the monomials

$$\bigoplus_{k=0}^{8} x_4^{8-k} x_5^k \ \oplus \ \bigoplus_{k=0}^{5} (x_2 \oplus x_3) x_4^{5-k} x_5^k \ \oplus \ \bigoplus_{k=0}^{2} (x_2^2 \oplus x_2 x_3 \oplus x_3^2) x_4^{2-k} x_5^k, \qquad (21)$$

all generic linear combinations of which fail to be transversal.

To construct the trans-polar of $\Delta^\star_{\mathscr{F}_3}$, we start by noting that the facets $\Phi_1, \cdots, \Phi_6 \subset \Delta^\star$ are all convex. Following Construction 3.1, we find

$$\Phi_1^\triangledown = \mu_1 = (-1, -1, -1), \qquad \Phi_2^\triangledown = \mu_2 = (-1, -1, 7), \qquad (22a)$$

$$\Phi_3^\triangledown = \mu_3 = (2, -1, -2), \qquad \Phi_4^\triangledown = \mu_4 = (2, -1, -1), \qquad (22b)$$

$$\Phi_5^\triangledown = \mu_5 = (-1, 2, -2), \qquad \Phi_6^\triangledown = \mu_6 = (-1, 2, -1). \qquad (22c)$$

The vertices $\mu_1, \mu_2$ indeed belong to $(\Delta^\star_{\mathscr{F}_3})^\circ$, but $\mu_3, \cdots, \mu_6$ lie beyond the fractional vertices of $(\Delta^\star_{\mathscr{F}_3})^\circ$: they delimit the *extension* (19), with the quadrangular facet $\Theta_1$ that lies in the $\nu_1^\circ = (x, 1{-}x, z)$ plane. Since $\nu_1 = [\nu_1, \nu_2] \cap [\nu_1, \nu_3] \cap [\nu_1, \nu_4] \cap [\nu_1, \nu_5]$, $\Theta_1 \subset \nu_1^\circ$ is more precisely delimited by the trans-polar images of the edges adjacent to $\nu_1$:

$$[\nu_1, \nu_2]^\triangledown = (-1, 2, z), \qquad [\nu_1, \nu_3]^\triangledown = (2, -1, z), \qquad (23a)$$

$$[\nu_1, \nu_4]^\triangledown = (x, 1{-}x, -1), \quad [\nu_1, \nu_5]^\triangledown = (x, 1{-}x, -2). \qquad (23b)$$

Following through in this fashion, we obtain the Newton polytope depicted on the right-hand side of Figure 3.

This result can be further corroborated as follows: With the (15)-standard "$\geqslant -1$" conditions, the computations (23) would have produced an empty set, since the standard polar operation (15) obscures the non-convexity of $\nu_1$. We may remedy this by "flipping" the defining inequalities in correlation with (non-)convexity: The extending facet $\Theta_1$ may be defined by first rewriting the $\langle \nu, u \rangle \geqslant -1$ condition in (15) as $(\langle \nu, u \rangle + 1) \geqslant 0$, and then flipping the sign of the left-hand side according to the (non-)convexity of $\nu$:

$$u \in M \otimes \mathbb{R}: \ \langle \nu_1, u \rangle = -1 \quad \& \quad \begin{cases} (-1)^F \big( \langle \nu_2, u \rangle + 1 \big) \geqslant 0, & F = 2; \\ (-1)^F \big( \langle \nu_3, u \rangle + 1 \big) \geqslant 0, & F = 2; \\ (-1)^F \big( \langle \nu_4, u \rangle + 1 \big) \geqslant 0, & F = 1; \\ (-1)^F \big( \langle \nu_5, u \rangle + 1 \big) \geqslant 0, & F = 1. \end{cases} \qquad (24)$$

Here $F = 1$ indicates that the usual condition for the polar ("$\geqslant -1$") is reversed—owing to the non-convexity of $\nu_1$ itself; the first two conditions ($F = 2$) are however flipped a second

---

[12]The "halo" on $\mathring{\mu}_i$ identifies them as fractional vertices of $(\Delta^\star)^\circ$.

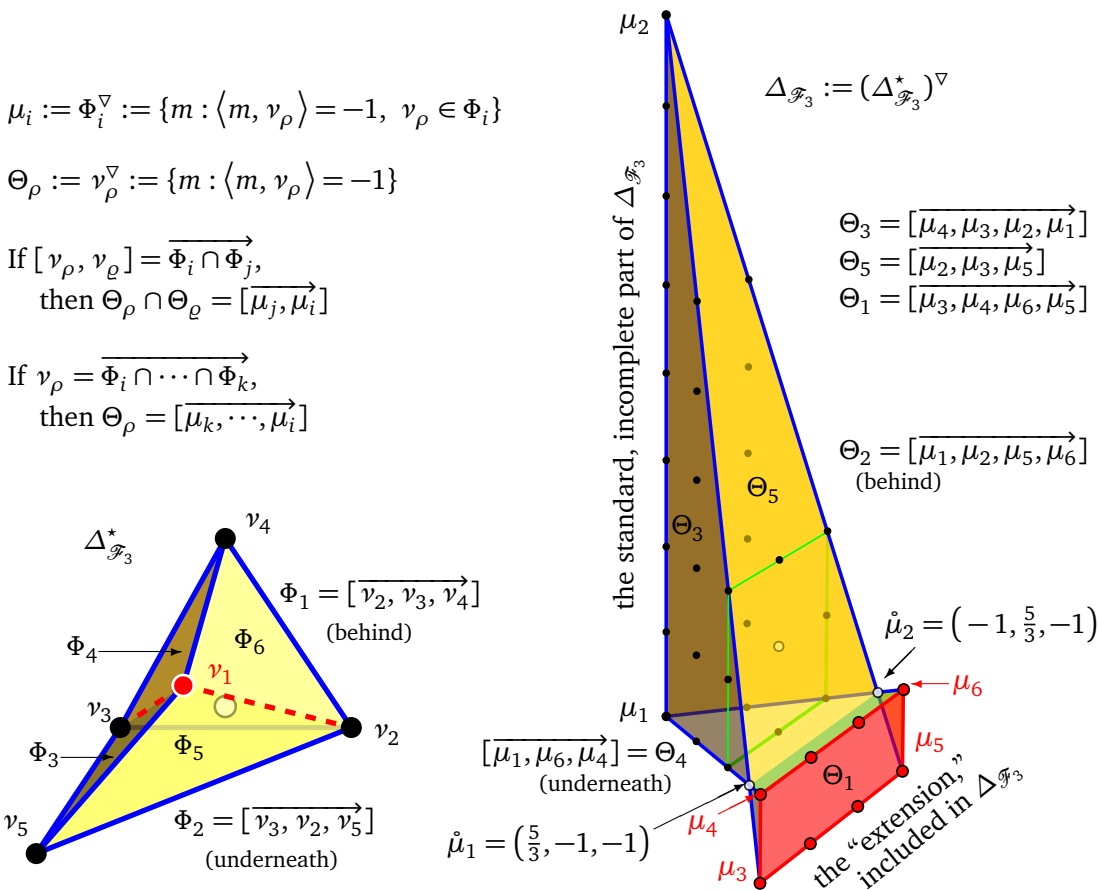

Figure 3: The spanning polytope $\Delta^{\star}_{\mathscr{F}_3}$ and the Newton polytope $\Delta_{\mathscr{F}_3}$ with several polar pairs of elements indicated. The outward orientation of $\Delta^{\star}_{\mathscr{F}_3}$ at a vertex $\nu_\rho$ orders the adjacent facets $\Phi_i \cap \cdots \cap \Phi_k$, and so induces the (*reverse*) ordering of $\Phi_i^{\triangledown}$, and the compatible orientation for $\Theta_\rho$.

time (and so back to the original inequality) owing to the fact that the edges $[\nu_1, \nu_2]$ and $[\nu_1, \nu_3]$ adjacent to $\nu_1$ are also non-convex. In passing, $[\nu_1, \nu_2, \nu_3]$ is the sub-polytope of the $\mathbb{P}^2$-fiber in $\mathscr{F}_m$. Proceeding in this way (corroborating *Step 3* of Construction 3.1) produces the complete Newton polytope $\Delta_{\mathscr{F}_3}$ shown to the right-hand side of Figure 3. In particular, the facets $\Theta_2, \Theta_3$ are self-crossing, owing to the fact that $\nu_2, \nu_3 \in \Delta^{\star}_{\mathscr{F}_3}$ are each adjacent to three convex and one non-convex edge.

The spanning polytope $\Delta^{\star}_{\mathscr{F}_3}$ admits a uniform outward orientation, which then induces an orientation of the Newton polytope $\Delta_{\mathscr{F}_3}$. At each vertex $\nu_\rho \in \Delta^{\star}_{\mathscr{F}_3}$, this orders the adjacent facets; for example, $\nu_5 = \overrightarrow{\Phi_5 \cap \Phi_3 \cap \Phi_2}$ implies[13] $\Theta_5 = [\overleftarrow{\mathring{\mu}_5, \mu_3, \mu_2}]$ — which gives the outward orientation to the "standard part" $[\overrightarrow{\mu_2, \mathring{\mu}_1, \mathring{\mu}_2}]$ of $\Theta_5$, but the *inward* orientation $[\overrightarrow{\mathring{\mu}_2, \mathring{\mu}_1, \mu_3, \mu_5}]$ within the extension. Similarly, $\nu_3 = \overrightarrow{\Phi_1 \cap \Phi_2 \cap \Phi_3 \cap \Phi_4}$ implies $\Theta_3 = [\overleftarrow{\mathring{\mu}_1, \mu_2, \mu_3, \mu_4}]$, giving the outward orientation to "standard part" $[\overrightarrow{\mu_1, \mathring{\mu}_1, \mu_2}]$ of $\Theta_3$, but the *inward* orientation $[\overrightarrow{\mu_4, \mu_3, \mathring{\mu}_1}]$ within the extension. In a similar fashion, this gives the facets $\Theta_4, \Theta_5$ and $\Theta_1$ the outward and opposite inward orientations, respectively. This orientation will be essential in the combinatorial formulae for the Euler and Hodge numbers, see section 4 and Appendix B.

---

[13]Duality relations preserve (reverse) orientations in even (odd) dimensions.

Finally, the standard formula (14) associates the Laurent monomials

$$[\mu_4, \mu_6]: \bigoplus_{k=0}^{3} \frac{x_2^{3-k} x_3^k}{x_4}, \qquad [\mu_3, \mu_5]: \bigoplus_{k=0}^{3} \frac{x_2^{3-k} x_3^k}{x_3}. \tag{25}$$

Straightforward computation shows that generic Laurent polynomials formed with both (21) and (25) are transversal (as discussed in Appendix A): following the GLSM analysis in Section 2, the polynomial $f(x)$ and its gradient $\partial f(x)/\partial x_i$ vanish simultaneously only in the Landau-Ginzburg orbifold phase, where $x_2, x_3, x_4, x_5 \to 0$.

## 4 Combinatorial calculations

Let us first consider Batyrev's Euler characteristic formula [21, 34]

$$\chi(\hat{Z}_f) = \sum_{k=1}^{\dim X - 2} (-1)^{k-1} \sum_{\substack{\dim(\theta)=k \\ \theta \subset \Delta_X}} d(\theta) d(\theta^*) \tag{26}$$

where $\hat{Z}_f$ is the MPCP desingularization of the anticanonical $\Delta_X$-regular (transversal) hypersurface $f(x) = 0$ in a toric variety $X$ and $\Delta_X$ is its Newton polytope. For every face $\theta \subset \Delta_X$ in the Newton polytope, $\theta^* \subset \Delta_X^\star$ is the dual face[14] in the spanning polytope such that $\dim(\theta) + \dim(\theta^*) = \dim(\hat{Z}_f)$. In fact, there is a natural generalization of (26) in which we sum over *all* the codimension $k$ faces,

$$\chi(\hat{Z}_f) = \sum_{k=-1}^{\dim X} (-1)^{k-1} \sum_{\substack{\dim(\theta)=k \\ \theta \subset \Delta_X}} d(\theta) d(\theta^*), \tag{27}$$

where for $\dim(\theta) = -1$, $\theta$ is the unique interior point in $\Delta_X$ and $\theta^* = \Delta_X^\star$, and similarly for $\dim(\theta) = \dim X$, where $\theta = \Delta_X$ and so $\theta^*$ is the unique interior point in $\Delta_X^\star$. Furthermore, because we restrict to star-triangulations, it follows that

$$d(\Delta_X^\star) = \sum_{\dim(\theta)=0} d(\theta) d(\theta^*), \quad d(\Delta_X) = \sum_{\dim(\theta)=\dim X - 1} d(\theta) d(\theta^*), \tag{28}$$

since $\dim(\theta^*) = \dim X - 1$ and $\dim(\theta^*) = 0$ and hence the two sums range over codimension-one faces in $\Delta$ and $\Delta^\star$, respectively. Thus, the contribution from $k = -1$ and $k = 0$ cancel, as do the $k = \dim X - 1$ and $k = \dim X$ terms, as expected for consistency.

We now will demonstrate that the trans-polar pair of polytopes $\left(\Delta_X^\star, \Delta_X := (\Delta_X^\star)^\nabla\right)$ provides for computing the basic topological characteristics — and that the orientations discussed between (21) and (25) turn out to be crucial. In light of this, we propose the following conjecture generalizing Batyrev's construction [21, 34]:[15]

**Conjecture 4.1.** *The Euler number, $\chi(\hat{Z}_f)$, and Hodge numbers, $h^{1,1}(\hat{Z}_f)$ and $h^{n-2,1}(\hat{Z}_f)$ for a Calabi-Yau $(n-1)$-fold $\hat{Z}_f$, which is the MPCP desingularization of the anticanonical $\Delta_X$-regular*

---

[14]For faces of all nonzero codimension, the dual of a face (as used here) is the trans-polar of that face and $\dim(\theta) + \dim(\theta^*) = n-1$. On the other hand, the dual of the entire polytope (codim $=0$) is not the trans-polar polytope (which has the same rather than the complementary dimension) but the unique integral point (the origin) that is internal to the entire trans-polar polytope.

[15]The toric construction of mirror pairs of Calabi-Yau manifolds restricted to Fermat hypersurfaces in weighted projective space was first observed by Roan [35].

(transversal) hypersurface $f(x) = 0$ in a toric variety $X$ with $\Delta_X$ its VEX Newton polytope, are given by (26),

$$h^{n-2,1}(\hat{Z}_f) = l(\Delta_X) - 5 - \sum_{\substack{\text{codim}\,\theta=1 \\ \Theta \subset \Delta_X}} l^*(\theta) + \sum_{\substack{\text{codim}\,\theta=2 \\ \theta \subset \Delta_X}} l^*(\theta) \cdot l^*(\theta^*), \tag{29}$$

and

$$h^{1,1}(\hat{Z}_f) = l(\Delta_X^\star) - 5 - \sum_{\substack{\text{codim}\,\theta^*=1 \\ \Phi \subset \Delta_X^\star}} l^*(\theta^*) + \sum_{\substack{\text{codim}\,\theta^*=2 \\ \phi \subset \Delta_X^\star}} l^*(\theta^*) \cdot l^*(\theta), \tag{30}$$

respectively, where $l(\theta)$ ($l(\theta^*)$) is the number of internal points in the face $\theta \subset \Delta_X$ ($\theta^* \subset \Delta_X^\star$), which may be negative if $\theta$ ($\theta^*$) has opposite orientation. The similarly constructed Calabi-Yau $(n{-}1)$-fold $\hat{Z}_g$, which is the MPCP desingularization of the anticanonical $\Delta_Y$-regular (transversal) hypersurface $g(y) = 0$ in a toric variety $Y$ with $\Delta_Y = \Delta_X^\star$ its VEX Newton polytope, is the mirror manifold to $\hat{Z}_f$, with the roles of $\Delta_X$ and $\Delta_X^\star$ interchanged, such that $h^{1,1}(\hat{Z}_g) = h^{n-2,1}(\hat{Z}_f)$ and $h^{n-2,1}(\hat{Z}_g) = h^{1,1}(\hat{Z}_g)$, and thus $\chi(\hat{Z}_g) = (-1)^{n-1}\chi(\hat{Z}_f)$.

We first focus on the example $K3 \subset \mathscr{F}_3$ from Section 3.2, followed by the corresponding Calabi-Yau threefolds, where we also calculate $h^{1,1}$ and $h^{2,1}$; additional examples may be found in the companion paper [24].

## 4.1 K3

Adapting (26) to the case of the $K3$, $\hat{Z}_f$ hypersurface $f(x) = 0 \subset \mathscr{F}_m$, the indicated summation should extend only over 1-dimensional faces (edges) $\theta \subset \Delta_{\mathscr{F}_m}$ in the Newton polytope (the dual of which, $\theta^* \subset \Delta_{\mathscr{F}_m}^\star$, are edges in the spanning polytope):

$$\chi(\hat{Z}_f) = \sum_{\substack{\dim(\theta)=1 \\ \theta \subset \Delta_{\mathscr{F}_m}}} d(\theta)\,d(\theta^*). \tag{31}$$

All edges $\theta^* \subset \Delta_{\mathscr{F}_m}^\star$ have unit degree for $m \neq 2$, in which case the sum reduces to the degrees of the edges in the Newton polytope $\Delta_{\mathscr{F}_m} = (\Delta_{\mathscr{F}_m}^\star)^\triangledown$. The $m \leqslant 2$ cases are well understood and convex so that the trans-polar operation reduces to the familiar polar (15). We then focus on $m \geqslant 3$.

As is evident from Figure 3 and 6 below, $\Delta_{\mathscr{F}_m}$ has a total of nine edges:

- the one tallest vertical edge $[\nu_2, \nu_3]^\triangledown = \Theta_2 \cap \Theta_3 = [\mu_1, \mu_2]$ has degree $2 + 2m$;
- two horizontal edges: $[\nu_2, \nu_4]^\triangledown = \Theta_2 \cap \Theta_4 = [\mu_1, \mu_6]$ and $[\nu_3, \nu_4]^\triangledown = \Theta_3 \cap \Theta_4 = [\mu_1, \mu_4]$, both of which have degree 3;
- two slanted edges: $[\nu_2, \nu_5]^\triangledown = \Theta_2 \cap \Theta_5 = [\mu_2, \mu_5]$ and $[\nu_3, \nu_5]^\triangledown = \Theta_3 \cap \Theta_5 = [\mu_2, \mu_3]$, both of which have degree 3;
- two horizontal edges in the extension: $[\nu_1, \nu_4]^\triangledown = \Theta_1 \cap \Theta_4 = [\mu_4, \mu_6]$ and $[\nu_1, \nu_5]^\triangledown = \Theta_1 \cap \Theta_5 = [\mu_3, \mu_5]$, both of which have degree 3;
- two vertical edges in the extension: $[\nu_1, \nu_2]^\triangledown = \Theta_1 \cap \Theta_2 = [\mu_5, \mu_6]$ and $[\nu_1, \nu_3]^\triangledown = \Theta_1 \cap \Theta_3 = [\mu_3, \mu_4]$, both of which have degree $-(m{-}2)$: a quick comparison with the $m \leqslant 2$ cases (see Figure 6) shows that for $m \geqslant 3$ these two edges manifestly extend in the direction opposite from the $m \leqslant 2$ cases.

Tallying these contributions produces in (31):

$$\chi(\hat{Z}_f) = (2+2m) + 2(3) + 2(3) + 2(3) + 2[-(m{-}2)] = 24, \quad m \geqslant 3. \tag{32}$$

In fact, the computation is also true for $m=2$; not only is the end result independent of $m$, but the method itself extends.

Finally, note that since (31) is completely symmetric in exchanging $\theta$ and $\theta^*$, it is clear that the mirror $K3$, $\hat{Z}_g$, to the hypersurface $f(x)=0 \subset \mathscr{F}_m$, is defined as a hypersurface $g(y)=0$ in the toric variety $Y$ constructed by exchanging the roles of $\Delta^*$ and $\Delta$.

## 4.2 Calabi-Yau three-folds

For illustration, consider the Calabi-Yau 3-fold $\hat{Z}_f$ hypersurfaces $f(x)=0$ in the Hirzebruch 4-folds $F_m$ ($m$-twisted $\mathbb{P}^3$-bundles over $\mathbb{P}^1$),[16] where in analogy with K3, $(x_1, \ldots, x_4)$ and $(x_5, x_6)$ are the homogeneous fiber and base coordinates for the $\mathbb{P}^3$ and $\mathbb{P}^1$, respectively.

| $\Delta^*_{F_m}$ | $v_0$ | $v_1$ | $v_2$ | $v_3$ | $v_4$ | $v_5$ | $v_6$ |
|---|---|---|---|---|---|---|---|
| | 0 | −1 | 1 | 0 | 0 | 0 | −m |
| fiber | 0 | −1 | 0 | 1 | 0 | 0 | −m |
| | 0 | −1 | 0 | 0 | 1 | 0 | −m |
| base | 0 | 0 | 0 | 0 | 0 | 1 | −1 |
| $\mathscr{M}(\Delta^*_{F_m})\{ \ell_1$ | −4 | 1 | 1 | 1 | 1 | 0 | 0 |
| $\ell_2$ | m−2 | −m | 0 | 0 | 0 | 1 | 1 |
| | $x_0$ | $x_1$ | $x_2$ | $x_3$ | $x_4$ | $x_5$ | $x_6$ |

(33)

For $n=4$, the extension in the Newton polytope, $(v_1)^\nabla$, is delimited by $[v_1, v_2]^\nabla$, $[v_1, v_3]^\nabla$, $[v_1, v_4]^\nabla$, $[v_1, v_5]^\nabla$ and $[v_1, v_6]^\nabla$, and so is the region defined by:

$$u \in M_{\mathbb{R}}: \ \langle v_1, u \rangle = -1 \quad \& \quad \begin{cases} (-1)^F \big( \langle v_2, u \rangle + 1 \big) \geqslant 0, & F=2; \\ (-1)^F \big( \langle v_3, u \rangle + 1 \big) \geqslant 0, & F=2; \\ (-1)^F \big( \langle v_4, u \rangle + 1 \big) \geqslant 0, & F=2; \\ (-1)^F \big( \langle v_5, u \rangle + 1 \big) \geqslant 0, & F=1; \\ (-1)^F \big( \langle v_6, u \rangle + 1 \big) \geqslant 0, & F=1. \end{cases}$$

(34)

Here $F=1$ indicates that the usual condition for the polar ("$\geqslant -1$") is reversed—again owing to the non-convexity of $v_1$; the first three conditions ($F=2$) are however flipped a second time (and so equal the original inequality) as the vertices $[v_1, v_2, v_3, v_4]$ form the sub-polytope of the $\mathbb{P}^3$-fiber in $F_m$; the 2-faces $[v_1, v_i, v_j]$ of this sub-polytope are non-convex in $\Delta^*_{F_m}$ for $m \geqslant 3$. In turn, $v_4$ and $v_5$ span the sub-polytope of the base $\mathbb{P}^1$.

The extension in the Newton polytope thus takes the form of a 3-sided prism,

$$\big\{ (x, y, 1-x-y, z), \ -1 \leqslant x \leqslant 3, \ -1 \leqslant y \leqslant (2-x), \ (1-m) \leqslant z \leqslant -1 \big\},$$

(35)

generalizing the rectangle for $n=3$ and vertical edge for $n=2$. The vertical edge remains the same for general $n$ while the base of the extension is an $(n-2)$-dimensional simplex.

**The Euler characteristic:** We first calculate the Euler number along the lines of the K3 in the previous subsection, evaluating the two terms in Batyrev's expression (26) for an ($n=4$)-dimensional ambient toric variety (see also [21, Theorem 4.5.3]):

$$\chi(\hat{Z}_f) = \sum_{\dim \theta = 1} d(\theta) d(\theta^*) - \sum_{\dim \theta = 2} d(\theta) d(\theta^*),$$

(36)

---

[16]The Hirzebruch 4-fold $F_m$ may also be described as a generic degree-$(1,m)$ hypersurface in $\mathbb{P}^4 \times \mathbb{P}^1$, in which the Calabi-Yau hypersurfaces have $h^{1,1}=2$, $h^{2,1}=86$, triple intersection numbers $\kappa_{1,1,1}=2+3m$ $\kappa_{1,1,2}=4$ and $\kappa_{1,2,2}=0=\kappa_{2,2,2}$ and the second Chern class evaluations $c_2[J_1]=44+6m$ and $c_2[J_2]=24$ [5]. Then, the integral basis change $(J_1, J_2) \to (J_1+cJ_2, J_2)$ identifies $F_m \xrightarrow{\approx} F_{m+4c}$ as diffeomorphic [36]. We verify that the toric specification (33) reproduces this (classical) topological data completely.

Here $\theta \subset \Delta_{F_m}$ are faces in the (extended) Newton polyhedron for $F_m$; $\theta^* \subset \Delta^\star_{F_m}$ are their trans-polar faces in the spanning polytope. Since now $\dim(\theta) + \dim(\theta^*) = \dim(\hat{Z}_f) = 3$, edges in the Newton polytope are paired with 2-faces in the spanning polytope and *vice versa*.

We refer the reader to the Appendix B for the details of the calculation. It suffices to say that in analogy with the $n = 3$ case of K3, there are in the extension of $\Delta_{F_m}$ three 1-faces $\theta$, i.e., edges in $\Delta_{F_m}$, for which $d(\theta) = -1$ and also three 2-faces $\theta$, i.e., edges in $\Delta_{F_m}$, for which $d(\theta) = -8$. In the latter, two of the four boundary 1-faces have $d(\theta) = -1$, while the other two have $d(\theta) = 4$. The result: $\chi = 56 - 224 = -168$, in agreement with the gCICY result [1, 5].

**The Hodge numbers:** We next turn to calculating the Hodge numbers $h^{2,1}$ and $h^{1,1}$ following Batyrev's formulae [21, Theorem 4.3.7]:

$$h^{2,1}(Y_m) = l(\Delta_{F_m}) - 5 - \sum_{\substack{\text{codim}\,\theta=1 \\ \Theta \subset \Delta_{F_m}}} l^*(\theta) + \sum_{\substack{\text{codim}\,\theta=2 \\ \theta \subset \Delta_{F_m}}} l^*(\theta) \cdot l^*(\theta^*), \tag{37}$$

and [21, Theorem 4.4.2]:

$$h^{1,1}(Y_m) = l(\Delta^\star_{F_m}) - 5 - \sum_{\substack{\text{codim}\,\theta^*=1 \\ \Phi \subset \Delta^\star_{F_m}}} l^*(\theta^*) + \sum_{\substack{\text{codim}\,\theta^*=2 \\ \phi \subset \Delta^\star_{F_m}}} l^*(\theta^*) \cdot l^*(\theta). \tag{38}$$

Here, $\theta^* \subset \Delta^\star_{F_m}$ is the facet dual to $\theta \subset \Delta_{F_m}$, and vice versa. To avoid the ambiguity of counting internal points in negative-degree faces, we rewrite $l^*(\theta)$ and $l^*(\theta^*)$ using the general formulae (53) and (51) to re-express the summands in terms of various $k$-face degrees. Thus, Batyrev's formulae (38) and (37) take the following form:

$$h^{2,1} = \sum_{\dim\theta=1} d(\theta) + \sum_{\dim\theta^*=1} d(\theta^*) - 4 + N_0 - N_1 + \tfrac{1}{2} \sum_{\dim\theta=2} \big(d(\theta) - c(\theta)\big) d(\theta^*), \tag{39}$$

$$h^{1,1} = \sum_{\dim\theta^*=1} d(\theta^*) + \sum_{\dim\theta=1} d(\theta) - 4 + N_0^* - N_1^* + \tfrac{1}{2} \sum_{\dim\theta^*=2} \big(d(\theta^*) - c(\theta^*)\big) d(\theta), \tag{40}$$

where $N_k$ ($N_k^*$) refers to the number of $k$-faces in $\Delta_{F_m}$ ($\Delta^\star_{F_m}$) and $c(\theta^{(2)})$ is the effective circumference of $\theta^{(2)}$, see (53). The reader can consult Appendix B.2 for the details of the calculation the result of which is that $h^{2,1} = 86$ and $h^{1,1} = 2$, independent of $m$ and as with the Euler number in agreement with the gCICY result [1, 5].

## 4.3 Mirror models

As mentioned above (20), the trans-polar pair of polytopes $(\Delta_{\mathscr{F}_m}, \Delta^\star_{\mathscr{F}_m})$ defines a pair of toric varieties, $(\mathscr{F}_m, \mathscr{F}_m^\triangledown)$, in which the (MPCP-desingularized) Calabi-Yau hypersurfaces are natural mirrors [21, 33]. In particular, the above explicit computations of the Euler and Hodge numbers verifies that swapping $\Delta_{\mathscr{F}_m} \leftrightarrow \Delta^\star_{\mathscr{F}_m}$ indeed has the expected mirror effect. We now explore this relationship further and provide additional corroboration to this relationship.

For concreteness, consider the trans-polar pair of polytopes $(\Delta_{\mathscr{F}_3}, \Delta^\star_{\mathscr{F}_3})$, see Figure 3. Expressed in terms of the homogeneous coordinates associated with the vertices of $\Delta^\star_{\mathscr{F}_3}$, we choose a minimal set of vertices of $\Delta_{\mathscr{F}_3}$ that will generate the $M$-lattice[17] [23]. That is, for the

---

[17]Restricting the standard part of the Newton polytope to $M$-integral points results in a *drastic* example of non-transversality: the origin is no longer internal but "surfaces" into the "cut-off" facet outlined in Figure 3. Since our inclusion of the "extension" (i.e., Laurent polynomials) restores transversality (as discussed in Appendix A) even in this drastic case, there most certainly exist much milder cases, where analogous "extensions" in the Newton polytope (and Laurent monomials) restore transversality.

example of $\mathscr{F}_3$, we select $\mu_1, \mu_2, \mu_3, \mu_5$ and obtain:

$$f_{\min}(\mathscr{F}_3) = a_1 x_5^8 + a_2 x_4^8 + a_3 \frac{x_2^3}{x_4} + a_5 \frac{x_3^3}{x_4} \qquad \in \mathbb{P}^3_{(3:3:1:1)}[8], \tag{41a}$$

$$g_{\min}(\mathscr{F}_3^{\triangledown}) = b_2 y_3^3 + b_3 y_5^3 + b_4 \frac{y_2^8}{y_3 y_5} + b_5 y_1^8 \quad \in \mathbb{P}^3_{(3:5:8:8)}[24], \tag{41b}$$

where the monomials in (41b) correspond to the vertices $\nu_\rho$ of the complex hull of $\Delta^\star_{\mathscr{F}_3}$, which analogously generate the $N$ lattice.[18] Straightforward computation along the lines of Appendix A shows that the generic polynomials (41) are transversal, due to the Laurent monomials, and the polynomials in each pair are each other's transpose:

$$(41a) \ \& \ (41b) \Rightarrow \quad \mathbb{M}_f(\mathscr{F}_3) = \begin{bmatrix} 0 & 0 & 0 & 8 \\ 0 & 0 & 8 & 0 \\ 3 & 0 & -1 & 0 \\ 0 & 3 & -1 & 0 \end{bmatrix} = \mathbb{M}_g^T(\mathscr{F}_3^{\triangledown}). \tag{42}$$

Following the prescription of Ref. [22], the maximal phase symmetry of (41a) is $\mathbb{Z}_3 \times \mathbb{Z}_8 \times \mathbb{Z}_{24}$, generated by $\gamma_1 := (\mathbb{Z}_3 : \frac{1}{3}, \frac{2}{3}, 0, 0)$, $\gamma_2 := (\mathbb{Z}_8 : 0, 0, 0, \frac{1}{8})$ and $\gamma_3 := (\mathbb{Z}_{24} : \frac{1}{24}, \frac{1}{24}, \frac{1}{8}, 0)$. Then, $9(\gamma_2 + \gamma_3) = (\mathbb{Z}_8 : \frac{3}{8}, \frac{3}{8}, \frac{1}{8}, \frac{1}{8})$ generates the "quantum symmetry," the discrete subgroup of the $\mathbb{P}^3_{(3:3:1:1)}$ projectivization, leaving a $\mathbb{Z}_3 \times \mathbb{Z}_{24}$ geometric symmetry generated for example as:

$$a_1 x_5^8 + a_2 x_4^8 + a_3 \frac{x_2^3}{x_4} + a_5 \frac{x_3^3}{x_4} : \quad \exp\left\{ 2i\pi \begin{bmatrix} \frac{1}{3} & \frac{2}{3} & 0 & 0 \\ \frac{1}{24} & \frac{1}{24} & \frac{1}{8} & 0 \\ \frac{3}{8} & \frac{3}{8} & \frac{1}{8} & \frac{1}{8} \end{bmatrix} \right\} \begin{bmatrix} x_2 \\ x_3 \\ x_4 \\ x_5 \end{bmatrix} : \quad \begin{cases} G = \mathbb{Z}_3 \times \mathbb{Z}_{24}, \\ Q = \mathbb{Z}_8. \end{cases} \tag{43}$$

Analogously, the maximal phase symmetry of (41b) is also $\mathbb{Z}_3 \times \mathbb{Z}_8 \times \mathbb{Z}_{24}$, generated by $\gamma_1^{\triangledown} := (\mathbb{Z}_3 : 0, 0, \frac{1}{3}, \frac{2}{3})$, $\gamma_2^{\triangledown} := (\mathbb{Z}_8 : \frac{1}{8}, 0, 0, 0)$ and $\gamma_3^{\triangledown} := (\mathbb{Z}_{24} : 0, \frac{1}{24}, \frac{2}{3}, \frac{2}{3})$. Then, $\gamma_2^{\triangledown} + 5\gamma_3^{\triangledown} = (\mathbb{Z}_{24} : \frac{1}{8}, \frac{5}{24}, \frac{1}{3}, \frac{1}{3})$ generates the "quantum symmetry," the discrete subgroup of the $\mathbb{P}^3_{(3:5:8:8)}$ projectivization, leaving a $\mathbb{Z}_3 \times \mathbb{Z}_8$ geometric symmetry generated for example as:

$$b_2 y_3^3 + b_3 y_5^3 + b_4 \frac{y_2^8}{y_3 y_5} + b_5 y_1^8 : \quad \exp\left\{ 2i\pi \begin{bmatrix} 0 & 0 & \frac{1}{3} & \frac{2}{3} \\ \frac{1}{8} & 0 & 0 & 0 \\ \frac{3}{24} & \frac{5}{24} & \frac{1}{3} & \frac{1}{3} \end{bmatrix} \right\} \begin{bmatrix} y_1 \\ y_2 \\ y_3 \\ y_5 \end{bmatrix} : \quad \begin{cases} G^{\triangledown} = \mathbb{Z}_3 \times \mathbb{Z}_8, \\ Q^{\triangledown} = \mathbb{Z}_{24}. \end{cases} \tag{44}$$

To swap the geometric and quantum symmetry, we should consider $\big((43), (44)/\mathbb{Z}_3\big)$ for a mirror pair of Landau-Ginzburg orbifold models, using the indicated $\mathbb{Z}_3$-action. In particular, upon this $\mathbb{Z}_3$-quotient, $\widetilde{G^{\triangledown}} = \mathbb{Z}_8 = Q$ and $\widetilde{Q^{\triangledown}} = \mathbb{Z}_3 \times \mathbb{Z}_{24} = G$.

We note that the ratio of the sizes of the geometric and the quantum symmetry groups equals the ratio of the degrees of the polytopes:[19]

$$\frac{|\widetilde{Q^{\triangledown}}|}{|\widetilde{G^{\triangledown}}|} = \frac{|G|}{|Q|} = \frac{3 \cdot 24}{8} = 9 = \frac{d(\Delta_{\mathscr{F}_3})}{d(\Delta^\star_{\mathscr{F}_3})} = \frac{54}{6} = \frac{d(\Delta^\star_{\mathscr{F}_3^{\triangledown}})}{d(\Delta_{\mathscr{F}_3^{\triangledown}})}. \tag{45}$$

---

[18]We note that there are two triangulations of the above minimal set of $\nu_\rho$ giving rise to two phases in the enlarged Kähler moduli space corresponding to $i)$ the orbifold phase of the hypersurface in the unresolved weighted projective space, and $ii)$ the Landau-Ginzburg orbifold, and similarly for the mirror geometry in terms of the $\mu_i$.

[19]Here the orders of the geometric and quantum symmetries are also computed by considering mirror interpretation of the $\mu_i$, as edges $\bar{\mu}_i = (\mu_i, 1)$ spanning a cone in $\bar{M} = M \oplus \mathbb{Z}$ describing the mirror geometry (and similarly for the $\nu_\rho$, with $\bar{\nu}_\rho = (\nu_\rho, 1)$ of a cone in $\bar{N} = N \oplus \mathbb{Z}$). The linearly independent $\bar{\mu}_i$ form a $4 \times 4$ matrix with determinant 72, describing the mirror toric ambient space as a discrete quotient of $\mathbb{C}^4$ of order $|G| = 72$. Similarly, the original toric variety is a $Q = \mathbb{Z}_8$ quotient of $\mathbb{C}^4$ since $\bar{\nu}_\rho$ form a $4 \times 4$ matrix with determinant 8 [37].

The analogous relationship persists also for any $n \geq 2$, and for $m \geq 0$. Given our choice of vertices and monomials (41), this chain of equalities is in complete agreement with the detailed analysis in Section 3 of Ref. [23]. The fact that these relationships continue to hold also for the $n = 2, 3, 4$ and $m \geq 0$ sequences of Laurent polynomials such as (41) we find to provide additional corroboration of our extension of toric methods in Section 3 and their application in Section 4, and of Construction 3.1 and Conjecture 4.1 in particular.

We close by noting that even without a detailed analysis of the Landau-Ginzburg orbifolds or the more complete mirror pair of GLSM's, the discrete symmetries are essential both in constructing the Hilbert spaces of the Landau-Ginzburg models and in significantly restricting the Yukawa couplings via the Wigner-Eckart theorem. In fact, the symmetries of the defining (Laurent) polynomials $f(x)$ and $g(y)$ will play such an important role in all phases of the corresponding GLSM's. We find it therefore significant that the phase symmetries fully conform to the transposition prescription [22, 23].

# 5 Conclusions and outlook

In this paper we have found that there is a natural generalization of the GLSM to Laurent superpotentials in which the phase-diagram of the enlarged Kähler moduli space, i.e., the secondary fan is constructed from the triangulations of the spanning polytope $\Delta_X^\star$ (and its convex hull) of non-Fano toric varieties. Our construction 3.1 specifies the "trans-polar" operation, which extends the standard "polar" operation so as to apply to all VEX polytopes, including the above $\Delta_X^\star$ and the Newton polyhedron $\Delta_X := (\Delta_X^\star)^\nabla$, as well as the original class of reflexive polytopes considered by Batyrev [21].

In particular, for polytopes corresponding to the Hirzebruch $n$-folds [5] $\mathscr{F}_m^{(n)}$ for $m \geq 0$ (collectively denoted in the subsequent listing as $X$), we have also shown that the spanning polytope $\Delta_X^\star$ and the Newton polytope $\Delta_X := (\Delta_X^\star)^\nabla$ admit a mutually compatible *orientation*. The orientation provides a sign for every face in each polytope and each cone in the fan that it spans. Furthermore, $\Delta_X^\star$ and $\Delta_X$ both admit oriented star-triangulations (compatible with the orientation of the polytopes), which provides a sign to the degree of every star-simplex. Allowing the degree of a star-simplex to be negative is crucial for correctly calculating the Euler and Hodge numbers from the combinatorial data in the pair of *oriented* polytopes $(\Delta_X^\star, \Delta_X)$ and their *oriented* star-triangulations by adapting and generalizing Batyrev's formula (26). Note that the *extension*, xtn$(\Delta_X)$, (19) within the (complete) Newton polytope $\Delta_X$ is essential not only for the computation of the above topological data, but also in that the Laurent monomials corresponding to the integral points of xtn$(\Delta_X)$ render the generic anticanonical polynomial transversal. Finally, just as $\Delta_X^\star$ spans the fan of $X$, $\Delta_{X^\nabla}^\star := \Delta_X$ defines a trans-polar toric $n$-fold $X^\nabla$, the fan of which is spanned by $\Delta_X$. Hence, swapping $\Delta_X^\star \leftrightarrow \Delta_{X^\nabla}^\star$ evidently induces the expected mirror effect on the Euler and Hodge numbers. This indicates the pair of (MPCP desingularized) anticanonical hypersurfaces in the toric varieties specified by the trans-polar pair of polytopes $(\Delta_X^\star, \Delta_X := \Delta_{X^\nabla}^\star)$ as prime candidates for mirror manifolds. Further evidence of mirror symmetry also follows from the exchange of "geometric" and "quantum" phase symmetries [22] for the Landau-Ginzburg orbifold phases obtained from a minimal choice of superpotentials, $f_{\min}(X)$ and $g_{\min}(Y)$, related by transposition of the matrix of exponents of anti-canonical monomials.

The present work has two natural extensions. First, it would be desirable to put the results presented herein on a rigorous mathematical footing, and in particular to prove Construction 3.1 and our main Conjecture 4.1, including to what extent they are valid. In their support, we have verified wherever possible, that the results reported herein are both self-consistent, and also consistent with the by now well established "generalized complete intersection" re-

sults [1, 5, 6]. Second, and the main motivation of the program at hand, is to understand the enlarged Kähler moduli space in order to calculate the non-perturbative corrections, i.e., Gromow-Witten invariants. This would allow us to determine whether the observed (mod $n$) periodicity within the class of Calabi-Yau hypersurfaces in the Hirzebruch $n$-folds is indeed broken by quantum corrections. This appears to be borne out by the cumulative effects determining the Kähler and complex structure discriminant loci [38], the self-consistency of which lends us hope that the present models have also a reasonable UV completion.

# Acknowledgements

We would like to thank Lara Anderson, Andreas Braun, Charles Doran, James Gray, Kevin Iga and Djordje Minić for helpful discussions on the topics in this article, as well as the anonymous referees for constructive suggestions. P.B. would like to thank the CERN Theory Group for their hospitality over the past several years. T.H. is grateful to the Department of Physics, University of Maryland, College Park MD, Department of Physics, University of Central Florida, Orlando FL and the Physics Department of the Faculty of Natural Sciences of the University of Novi Sad, Serbia, for the recurring hospitality and resources.

**Funding information**    The work of P.B. was partially supported by the National Science Foundation grant PHY-1207895 as well as by a Scientific Associateship position at CERN. Any opinions, findings, and conclusions or recommendations expressed in this material are those of the author(s) and do not necessarily reflect the views of the National Science Foundation.

# A   Transversality of $f(x)$

The $F$-terms in the GLSM potential (1a) consists of the modulus-squares of the derivatives of the superpotential, $\sum_i |\partial_i W(x)|^2$, which must vanish in the ground state (5a)–(5d). By quasi-homogeneity, these equations imply $W(x) = 0$, so that the ground state of the GLSM is by definition located within the base-locus of the superpotential, $\{\partial_i W(x) = 0 = W(x)\}$, further restricted by the mixed (5e) and $D$-term conditions (6). Since $W(x) = x_0 \cdot f(x)$, the "geometric" component $\{x_0 = 0 = f(x)\}$ is enforced if $f(x)$ is *transversal*, i.e., when the base-locus $\{f(x) = 0 = \partial_i f(x)\}$ of $f(x)$ is within the Stanley-Reisner ideal, excised from this component. Complementarily, the "non-geometric" component (with $x_0 \neq 0$) is then forced precisely to the base-locus of $f(x)$. Although GLSMs with non-transversal defining functions have long since also been considered [23, 39] and have some fascinating characteristics [40], we defer such generalizations for now.

   Herein, we adapt these standard notions of base-locus, transversality (and related $\Delta$-regularity [21]) to Laurent polynomials — and expressly so as to agree with the analogous (and variously confirmed) "generalized complete intersections" results [1,5], for which Ref. [6] provides a rigorous, scheme-theoretic formulation within the Čech cohomology framework. We expect a similarly rigorous formulation of the toric hypersurfaces discussed herein to be just as viable, but for this "proof-of-concept" note rely on the "working definition" motivated and discussed below. Wherever possible, we verify that the so-obtained results are fully self-consistent, as well as consistent with the by now well established "generalized complete intersection" results [1, 5, 6].

   We start with the observation that an algebraic sub-variety $X \subset A$ is defined as the zero-locus, $X = f^{-1}(0)$, of a *section* of a bundle (or sheaf) over $A$, which does not have to have a globally well-defined (regular, holomorphic) representative in any particular coordinate ring—

even before passing to their coordinate reparametrization equivalence classes. This is particularly manifest in Refs. [1,5], where Calabi-Yau varieties have been constructed by means of a sequence of embeddings:

$$\Big(X = \{x \in F : f(x) = 0\}\Big) \hookrightarrow \Big(F = \{x \in A : p(x) = 0\}\Big) \hookrightarrow A, \tag{46}$$

and where $f(x) \in H^0(F, \mathcal{K}_F^*)$ so $c_1(X) = 0$. Of interest are cases where $F \subset A$ is not Fano, so that (the pull-back to $A$ by $p$ of) $\mathcal{K}_F^*$ is *negative* over a factor in $A$, typically a $\mathbb{P}^1$. Then, (the pull-back to $A$ by $p$ of) $f(x)$ must have poles at certain points in $A$. In all cases of interest [1,5,6] however, the section $f(x)$ may be "tuned" so that the equivalence class $[f(x) + \lambda(x) \cdot p(x)]$ contains a well-defined (albeit *local*) representative at every point of $A$, "dialed" by suitable choices of $\lambda(x)$; see also Ref. [24]. These various representatives are all identical as sections on $F$, where $p(x) = 0$ by definition.

We now turn to the the present constructions where $F$ is itself a toric variety (rather than a hypersurface in some toric variety $A$), and compare results with Refs. [1,5] whenever possible, verifying that those results do not depend on the representation and/or embedding.

We recall Batyrev's "regularity conditions for hypersurfaces" [21, §3.1]: Given a polytope $\Delta \subset M_{\mathbb{R}} \subseteq \mathbb{R}^n$ and a corresponding polynomial $f(\xi)$ in $\xi \in \mathbb{C}^n$, (1) consider the non-compact zero-locus $Z_{f,\Delta} = \{\xi \in (\mathbb{C}^*)^n | f(\xi) = 0\}$, then (2) define $\bar{Z}_{f,\Delta}$ to be the closure of $Z_{f,\Delta} \subset (\mathbb{C}^*)^n$ in $\mathbb{P}_\Delta$; finally, (3) extend this definition to rational polyhedral fans $\Sigma \subseteq \mathbb{R}^n$.

It is the definition of (toric) closure $Z_{f,\Delta} \to \bar{Z}_{f,\Delta}$, in Batyrev's second step, that must be amended for rational monomials (corresponding to extensions in VEX polytopes $\Delta$) — both for the defining polynomial $f(x)$ but also for its derivatives as they appear in the standard definition of the base locus, i.e., the GLSM ground state (5). For a concrete example, consider the $(n, m) = (2, 3)$ case of (2):

$$f(x) = a_{21}\frac{x_2^2}{x_3} + a_{22}\frac{x_2^2}{x_4} + a_1 x_1^2 x_3^5 + a_2 x_1^2 x_4^5 = 0, \tag{47a}$$

$$\partial_1 f(x) = 2x_1(a_1 x_3^5 + a_2 x_4^5) = 0, \qquad \partial_2 f(x) = 2x_2\Big(\frac{a_{21}}{x_3} + \frac{a_{22}}{x_4}\Big) = 0, \tag{47b}$$

$$\partial_3 f(x) = 5a_1 x_1^2 x_3^4 - a_{21}\frac{x_2^2}{x_3^2} = 0, \qquad \partial_4 f(x) = 5a_2 x_1^2 x_4^4 - a_{22}\frac{x_2^2}{x_4^2} = 0. \tag{47c}$$

To define the zero-locus $\bar{Z}_f$ of any Laurent polynomial $f(x)$, we: (1) find the (incomplete) zero-locus $Z_f$ by omitting the (putative) pole-set, then (2) complete the zero-locus to $\bar{Z}_f$ by including the "intrinsic limits," where the limiting process is restricted to $Z_f$. Proceeding in this fashion for each of the polynomials in a system such as (47), we find the common zero-locus, $\bar{Z}_f \cap \bigcap_{i=1}^4 \bar{Z}_{\partial_i f}$.

For example, for $x_3, x_4 \neq 0$, (47a) yields $x_2 = \pm i\sqrt{\frac{x_1^2 x_3 x_4 (a_1 x_3^5 + a_2 x_4^5)}{a_{22} x_3 + a_{21} x_4}}$, defining the $x_3, x_4 \neq 0$ incomplete zero-locus, $Z_f$. This solution defines the "intrinsic limit" to the (putative) pole $x_3 \to 0$, where $x_2 = O(\sqrt{x_3}) \to 0$, preserving $f(x) = 0$ by definition; the $x_4 \to 0$ intrinsic limit is analogous. This adds the "intrinsic limit" points $(x_1, 0, 0, x_4)$ and $(x_1, 0, x_3, 0)$, completing $Z_f \to \bar{Z}_f$. As long as the (putative) pole-set $P_f$ and the (intrinsically completed) zero-locus $\bar{Z}_f$ have normal crossings, each point of $P_f \cap \bar{Z}_f$ has an open punctured neighborhood that is entirely within $Z_f$, and the above "$f$-intrinsic limit" should be well defined point-by-point. Heuristically, the vanishing of each of the polynomials in a system such as (47) forces the limits to the (putative) pole-locations such as $x_3 \to 0$ to be balanced by a correlated vanishing of the numerator, effectively keeping the rational monomials from diverging.

In practice, solving an algebraic system such as (47) is equivalent to "clearing the denominators,"[20] and is in this respect very similar to the practical computations in the Čech

---

[20]We thank S. Katz and D. Morrison for discussions on this point.

cohomology framework of Ref. [6]; see also Ref. [24]. In particular, the scheme-theoretic setting should afford a straightforward and systematic separation of the zero-locus from the pole-locus. This suggests that a similarly rigorous definition of the complete zero-locus of Laurent algebraic systems such as (47) is possible, but such a rigorous (re)definition is clearly beyond our present scope.

—⋆—

In this fashion, the base-locus of the system (47) is found to be $(0, 0, x_3, x_4) \cup (x_1, 0, 0, 0)$: The equations (47) hold simultaneously in the two intrinsic limits identified in Table 1:

1. $x_1, x_2 \to 0$, and with $(x_3, x_4)$ free; this is phase IV, including its boundaries *i* and *iv*.

2. $x_2, x_3, x_4 \to 0$, provided (47c) are solved by setting $x_3 = \alpha \sqrt[3]{\frac{x_2}{x_1}}$ and $x_4 = \beta \sqrt[3]{\frac{x_2}{x_1}}$ with $\alpha^6 = \frac{a_{21}}{5a_1}$ and $\beta^6 = \frac{a_{22}}{5a_2}$, whereupon the $x_2 \to 0$ may be taken requiring only that $x_2 < x_1$ but leaving $x_1$ otherwise free; this is phase III including its boundaries *iii* and *iv*.

Including other monomials from the Newton polytope makes the system more generic, and is on general grounds expected not to worsen the above behavior.

Note that the GLSM analysis in Section 2 is more detailed as it catalogues the branches of the base-locus of the superpotential $x_0 \cdot f(x)$, not just $f(x)$.

# B    Combinatorial calculations

## B.1    K3

Adapting again Batyrev's $n \geq 4$ formula [21,34], we write:

$$h^{1,1} = \Big[ l(\Delta^\star_{\mathscr{F}_m}) - 4 - \sum_{\substack{\mathrm{codim}(\theta)=1 \\ \theta \subset \Delta^\star_{\mathscr{F}_m}}} l^*(\theta) \Big] + \Big[ l(\Delta_{\mathscr{F}_m}) - 4 - \sum_{\substack{\mathrm{codim}(\theta)=1 \\ \theta \subset \Delta_{\mathscr{F}_m}}} l^*(\theta) \Big] + \Big[ \sum_{\substack{\mathrm{codim}(\theta)=2 \\ \theta \subset \Delta^\star_{\mathscr{F}_m}}} l^*(\theta) l^*(\theta^*) \Big], \quad (48)$$

where the first part is the contribution from the Picard group, the second counts the "toric" deformations of the complex structure, and the final term is a correction term which can be thought of as counting either non-polynomial deformations of the complex structure or non-toric Kähler deformations.

We now address these terms in turn.

**Picard term:**    It should be manifest from Figure 3 that:[21]

$$l(\Delta^\star_{\mathscr{F}_m}) = 6 \qquad \text{and} \qquad l^*(\Phi_i) = 0 \ \text{ for all } \ \Phi_i \subset \Delta^\star_{\mathscr{F}_m} \qquad (49)$$

independently of $m$: the constellation of $v_1, v_2, v_3, v_4$ remains fixed for all $m \geqslant 0$, and only $v_5 = (-m, -m, -1)$ is moved further and further into the 7th octant. However, as $m$ grows, neither the facets $\Phi_2, \Phi_3, \Phi_5$ nor the edges $[v_2, v_5], [v_3, v_5], [v_1, v_5]$ acquire any internal points. Therefore,

$$\Big[ l(\Delta^\star_{\mathscr{F}_m}) - 4 - \sum_{i=1}^{6} l^*(\Phi_i) \Big] = [6 - 4 - 6 \cdot (0)] = 2. \qquad (50)$$

This result perfectly agrees with the homology algebra computation: $H^2(\mathscr{F}_m, \mathbb{Z})$ is indeed 2-dimensional, generated by the Kähler forms of $\mathbb{P}^3 \times \mathbb{P}^1$ when realizing $\mathscr{F}_m$ as a degree-$(1, m)$ hypersurface in $\mathbb{P}^3 \times \mathbb{P}^1$, and both generators are for all $m \geqslant 0$ inherited by the Calabi-Yau hypersurface $K3 \subset \mathscr{F}_m$ [5].

---

[21]The fact that $l(\Delta^\star_{\mathscr{F}_m}) = \chi(\mathscr{F}_m)$ generalizes Corollary 7.3 in [34] to our class of non-Fano toric varieties constructed in terms of the non-reflexive $\Delta^\star_{\mathscr{F}_m}$, as well as to arbitrary dimension $n$ of the ambient toric variety.

**Toric deformations:** In order to calculate the second (and third) term in (48) it is necessary to understand the contribution of the extension part of the Newton polytope $\Delta_{\mathscr{F}_m}$ and how it varies with $m$. In particular we need to understand how to count the effective number of integral points interior to the various codimension one and two faces. Since $\Theta_1$ and portions of $\Theta_2$ and $\Theta_3$ are negatively oriented (see Section 3.2), this will require some care.

We find it expedient to relate the number of points in the relative interior of a face to the degree of that face:

**0-dim.:** $d(\theta^{(0)}) = 0! \cdot \text{Vol}_0(\theta^{(0)}) = 1$ for $\theta^{(0)}$ a vertex.

**1-dim.:** $d(\theta^{(1)}) = 1! \cdot \text{Vol}_1(\theta^{(1)})$ for an edge $\theta^{(1)}$ of length $\text{Vol}_1(\theta^{(1)})$. Since every edge has two end-points, and is subdivided by its $l^*(\theta^{(1)})$ interior points into $l^*(\theta^{(1)}) + 1$ unit-length 1-simplices, it follows that

$$l^*(\theta^{(1)}) = d(\theta^{(1)}) - 1. \tag{51}$$

Note that a negatively oriented edge of (signed) length $-\ell$ (formally) therefore has $-(\ell + 1)$ points in its relative interior; a negatively oriented unit-size edge (formally) has $-2$ points in its relative interior.

**2-dim.:** $d(\theta^{(2)}) = 2! \cdot \text{Vol}_2(\theta^{(2)})$ for a 2-face $\theta^{(2)}$ of area $\text{Vol}_2(\theta^{(2)})$. Let $\theta^{(2)}$ denote a 2-face that has $l^*(\theta^{(2)})$ (integral) points in its relative interior, $N$ vertices $\theta_i^{(0)}$ (and so also $N$ edges $\theta_i^{(1)}$), and $l^*(\theta_i^{(1)})$ (integral) points in the relative interior of the $i^{\text{th}}$ edge $\theta_i^{(1)}$. Thereupon, the degree of $\theta^{(2)}$ is obtained by counting the number of unit-area 2-simplices:[22]

$$d(\theta^{(2)}) = \sum_i d(\theta_i^{(0)}) - 2 + \sum_i l^*(\theta_i^{(1)}) + 2l^*(\theta^{(2)}). \tag{52}$$

Solving for $l^*(\theta^{(2)})$, using (51) and that $\sum_i d(\theta_i^{(0)}) = N_1$, we have Pick's theorem [41]:

$$l^*(\theta^{(2)}) = \tfrac{1}{2}\big[\, d(\theta^{(2)}) + 2 - c(\theta^{(2)})\big], \quad c(\theta^{(2)}) = \sum_{\theta_i^{(1)} \subset \partial\theta^{(2)}} d(\theta_i^{(1)}), \tag{53}$$

with the sum ranging over the boundary 1-faces of $\theta^{(2)}$, some of which may be negatively oriented and so contribute negatively. Using (53), we rewrite

$$\Big[l(\Delta_{\mathscr{F}_m}) - 4 - \sum_{\rho=1}^{5} l^*(\Theta_\rho)\Big] = \Big[l(\Delta_{\mathscr{F}_m}) - 4 - \sum_{\rho=1}^{5} \tfrac{1}{2}\big[d(\Theta_\rho) + 2 - c(\Theta_\rho)\big]\Big]. \tag{54}$$

To this end, we need the degrees of the facets, $d(\Theta_\rho)$, as well as the degrees of all the edges of each facet to calculate $c(\Theta_\rho)$. For example, $\Theta_2, \Theta_3 \subset \Delta_{\mathscr{F}_3}$ are self-crossing; see Figure 4 for a depiction of the (coarse) star-simplex over $\Theta_3$, shown from two points of view; the cone $\measuredangle(\Theta_3)$ (and $\measuredangle(\Theta_2)$ similarly) is self-crossing — and certainly unusual. Since $\Theta_3$ (and $\Theta_2$) is a quadrangle, it can be subdivided into two (coarse) simplices: one with a positive (CCW) orientation, the other with a negative (CW) orientation; see the right-hand diagram in Figure 4. These simplicial bases overlap and partially cancel, so as to reproduce the original, self-crossing facet. Similarly, the circumference of $\Theta_3$ is calculated by summing over the edges of $\Theta_3$ taking

---

[22]We thank K. Iga for independently verifying this result.

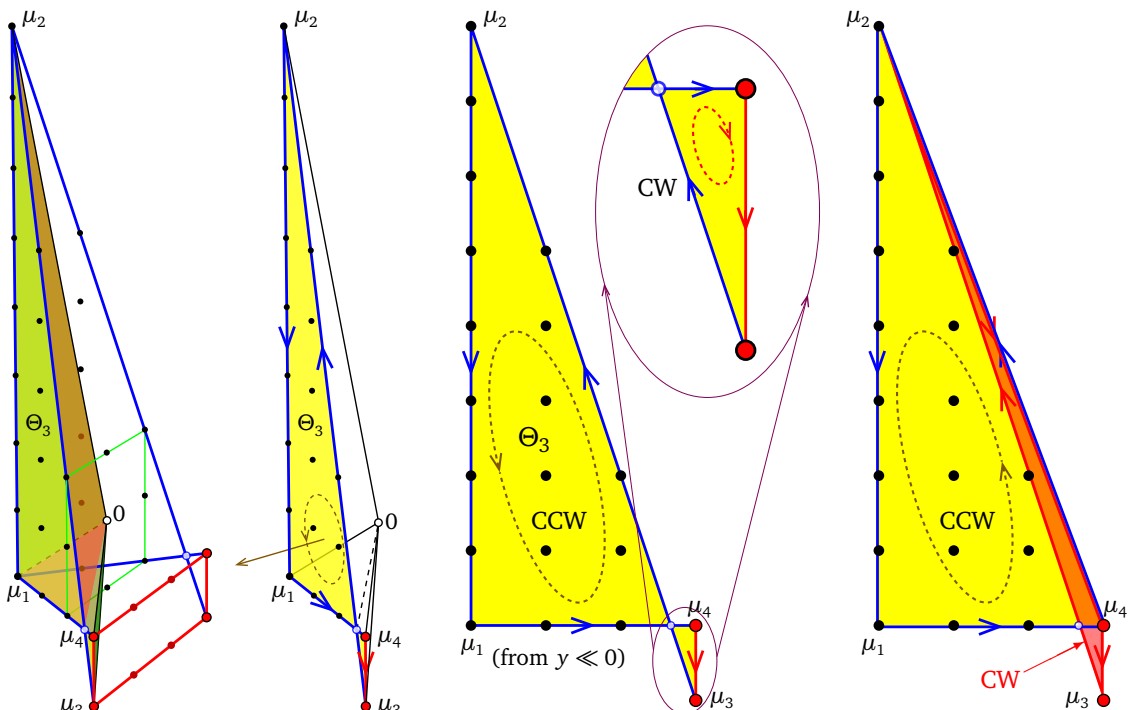

Figure 4: The star-simplex over the flip-folded facet $\Theta_3 \subset \Delta_{\mathscr{F}_3}$: in location within $\Delta_{\mathscr{F}_3}$ (left); the outward orientated "proper portion" (mid-left); $\Theta_3$ viewed from $y \ll 0$ and zooming on the "extension part" (mid-right); $\Theta_3$ subdivided into a CCW- and a CW-oriented coarse simplex (right)

into account that[23] $d([\overrightarrow{\mu_4, \mu_3}]) = 2-m$. Thus,

$$d(\Theta_3) = d([\overrightarrow{\mu_1, \mu_4, \mu_2}]) + d([\overrightarrow{\mu_2, \mu_4, \mu_3}]) = 3(2+2m) + 3(2-m) = 12 + 3m, \tag{55a}$$

$$c(\Theta_3) = d([\overrightarrow{\mu_1, \mu_4}]) + d([\overrightarrow{\mu_4, \mu_3}]) + d([\overrightarrow{\mu_3, \mu_2}]) + d([\overrightarrow{\mu_2, \mu_1}]),$$
$$= 3 + (2-m) + 3 + (2+2m) = 10 + m. \tag{55b}$$

and similarly for the other facets of $\Delta_{\mathscr{F}_m}$.

This now allows us to calculate the number of interior points in the codimension one faces of $\Delta_{\mathscr{F}_m}$ using (53).

$\Theta_2, \Theta_3$: From (55) and (53), $l^*(\Theta_2) = l^*(\Theta_3) = 2 + m$.

$\Theta_4, \Theta_5$: From Figure 5 we read off $d(\Theta_4) = 9 = d(\Theta_5)$ and $c(\Theta_4) = 9 = c(\Theta_5)$, which gives $l^*(\Theta_4) = l^*(\Theta_5) = 1$.

$\Theta_1$: From Figure 5 we read off $d(\Theta_1) = 3(2-m)2$ as well as $c(\Theta_1) = 2(3+(2-m)) = 10-2m$, which gives $l^*(\Theta_1) = 2 - 2m$.

Putting these together, we find:

$$\sum_{\substack{\text{codim}(\theta)=1 \\ \theta \subset \Delta_{\mathscr{F}_m}}} l^*(\theta) = \left[ 2 \cdot (2+m) + 2 \cdot 1 + (2-2m) \right] = 8. \tag{56}$$

and thus $\dim(\text{Aut}(\mathscr{F}_m)) = 4 + 8 = 12$, independent of $m$.

It remains to find $l(\Delta_{\mathscr{F}_m})$, the effective number of integral points in $\Delta_{\mathscr{F}_m} = (\Delta^\star_{\mathscr{F}_m})^\nabla$, which

---

[23]The notation $\Theta = [\theta_1, \cdots, \theta_k]$ is intended as a generalization of an edge: the faces $\theta_1, \cdots, \theta_k$ delimit $\Theta$ and are in its boundary. For example, $\Theta = [\mu_1, \cdots, \mu_\rho]$ states that "the vertices $\mu_1, \cdots, \mu_\rho$ span (and delimit) the face

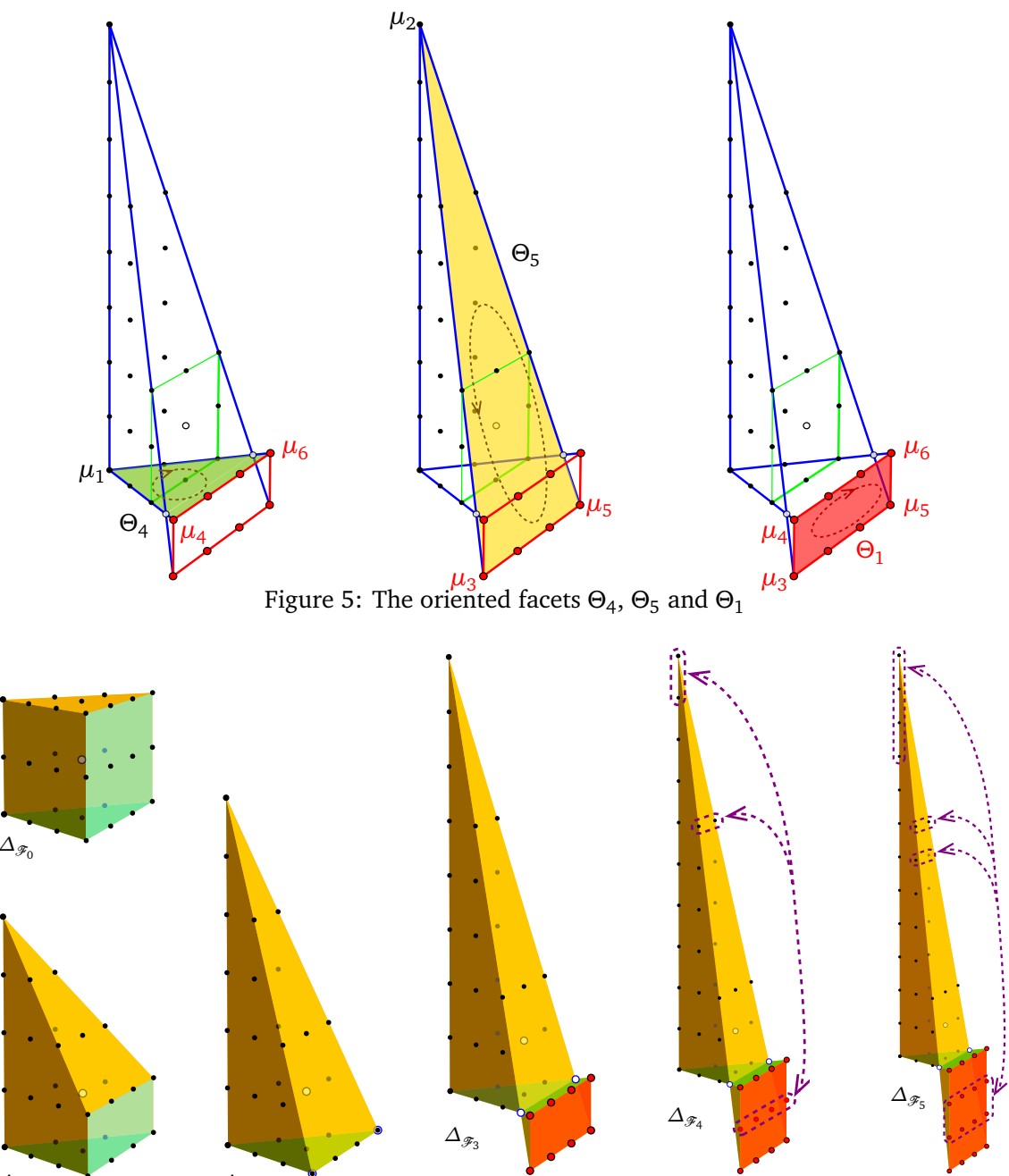

Figure 5: The oriented facets $\Theta_4$, $\Theta_5$ and $\Theta_1$

Figure 6: The complete Newton polytopes of the first six Hirzebruch 3-folds; see text

turns out to be $m$-independent and totals 30 (see Figure 6). To see this, we can proceed in two ways:

Rewrite $l(\Delta_{\mathscr{F}_m})$ by summing over all faces as follows:

$$l(\Delta_{\mathscr{F}_m}) = 1 + \sum_{\substack{\mathrm{codim}(\theta)=1 \\ \theta \subset \Delta_{\mathscr{F}_m}}} l^*(\theta) + \sum_{\substack{\mathrm{codim}(\theta)=2 \\ \theta \subset \Delta_{\mathscr{F}_m}}} l^*(\theta) + N_0, \qquad (57)$$

where $N_0 = 6$ is the number of vertices in $\Delta_{\mathscr{F}_m}$. By analyzing the Newton polytopes

_____________________

$\Theta$". The last step of Construction 3.1 relies on the inclusion-reversing nature of all duality relations, so that: (**1**) if $\Theta = \theta_1 \cap \cdots \cap \theta_k$ then $\Theta^\triangledown = [\theta_1^\triangledown, \cdots, \theta_k^\triangledown]$, i.e., $\theta_1^\triangledown, \cdots, \theta_k^\triangledown$ delimit $\Theta^\triangledown$; (**2**) if $\Theta = [\theta_1, \cdots, \theta_k]$ then $\Theta^\triangledown = \theta_1^\triangledown \cap \cdots \cap \theta_k^\triangledown$. Since VEX polytopes need not be convex, neither are their faces to be assumed convex, so that "to be spanned by" is not synonymous to "a convex linear combination."

$\Delta_{\mathscr{F}_0}, \cdots, \Delta_{\mathscr{F}_5}$ in Figure 6 we find that

$$\sum_{\substack{\mathrm{codim}(\theta)=2 \\ \theta \subset \Delta_{\mathscr{F}_m}}} l^*(\theta) = 1 + 2m + 6 \cdot 2 + 2(1-m) = 15. \tag{58}$$

Thus, it follows that the effective number of points in the (complete) Newton polytope is

$$l(\Delta_{\mathscr{F}_m}) = 1 + 8 + 15 + 6 = 30. \tag{59}$$

Alternatively, we can simply count the number of points in $\Delta_{\mathscr{F}_m}$:

1. The "standard part of $\Delta_{\mathscr{F}_m} \leftrightarrow H^0(\mathscr{F}_m, \mathscr{K}^*)$ has[24] $30 + \vartheta_3^m \cdot 4(m-3)$ integral points. This is in perfect agreement with the result of standard homological algebra [5, Eq. (A.28)].
2. The $m \geqslant 3$ extensions ($\Theta_1$, the red (dark-shaded) quadrangles in Figure 6) consist of two parts:
   (a) The top and bottom edge of $\Theta_1$ are $\Theta_1 \cap \Theta_4$ and $\Theta_1 \cap \Theta_5$, respectively. Those integral points count positively as belonging to $\Theta_4, \Theta_5$, but negatively as belonging to the negative-degree $\Theta_1$, and so cancel out.
   (b) The outlined $4(m-3)$ integral points within the (negative-degree) $\Theta_1$, including the ones on the side that are shared with the flip-oriented edge of the self-crossing $\Theta_2$ and $\Theta_3$.
   These latter $4(m-3)$ integral points contribute negatively and precisely cancel the excess integral points in the rising tip of the standard part of $\Delta_{\mathscr{F}_m}$.

The net result is that the *oriented* Newton polytope $\Delta_{\mathscr{F}_m} := (\Delta_{\mathscr{F}_m}^\star)^\triangledown$ encodes an effective number of 30 elements of $H^0(\mathscr{F}_m, \mathscr{K}^*)$. Summarizing the calculation, the contributions from the growing "tip" of the positively oriented portion of the Newton polytope and the growing negatively oriented extension cancel in just the same way also for Calabi-Yau hypersurfaces in Hirzebruch 2- and 4-folds [24]. Thus, the number of toric deformations is

$$\left[ l(\Delta_{\mathscr{F}_m}) - 4 - \sum_{\rho=1}^{5} l^*(\Theta_\rho) \right] = \left[ 30 - 4 - 8 \right] = 18. \tag{60}$$

**Correction term:** Finally, the "correction term," $\sum_\theta l^*(\theta) l^*(\theta^*)$ ranging over codimension-2 faces $\theta \subset \Delta_{\mathscr{F}_m}$, identically vanishes. To see this, note that $\theta^*$ are edges in the spanning polytope $\Delta_{\mathscr{F}_m}^\star$, all of which have positive unit degree, and no internal points by (51); with all $l^*(\theta^*) = 0$, the sum vanishes.

Putting (50), (60) and zero for the third, "correction" term in (48), we obtain:

$$h^{1,1}(K3 \subset \mathscr{F}_m) = 2 + 18 + 0 = 20, \tag{61}$$

as expected for a $K3$ surface.

## B.2 Calabi-Yau three-folds

We now turn to calculating the degrees of the various faces in the pair of polytopes $(\Delta_{F_m}, \Delta_{F_m}^\star)$, for $m \geq 3$.

---

[24]The symbol $\vartheta_x^y := 1$ if $x \leqslant y$ and $\vartheta_x^y := 0$ if $x > y$ is the usual step-function.

**dim** $\theta, \theta^* = 1$: The edges $\theta^* = \theta^\triangledown$ in the spanning polytope $\Delta^\star_{F_m}$ all have unit length so unit degree, $d(\theta^*) = 1$. Turning next to the edges $\theta$ in the Newton polytope $\Delta_{F_m}$ we have:

1. one "vertical" edge, $[(-1,-1,-1,-1),(-1,-1,-1,1+3m)]$, of degree $d(\theta)=2+3m$;
2. three "horizontal" edges of degree $d(\theta) = 4$ each, from $(-1,-1,-1,-1)$ to one of

$$\text{extension, top}: \quad (3,-1,-1,-1), \quad (-1,3,-1,-1), \quad (-1,-1,3,-1); \qquad (62)$$

3. three "slanted" edges of degree $d(\theta) = 4$ each, from $(-1,-1,-1,1+3m)$ to one of

$$\text{extension, bottom}: \quad (3,-1,-1,1-m), \quad (-1,3,-1,1-m), \quad (-1,-1,3,1-m). \quad (63)$$

4. In the extension part of the Newton polyhedron we have two sets of three "horizontal" edges, connecting two of three vertices on the top (62), and two of the three vertices on the bottom (63), each edge of degree $d(\theta) = 4$.
5. Finally, there are the three "vertical" negative-degree edges, each of which connects one of the three top vertices (62) to the corresponding one of the three bottom vertices (63). Each of these "vertical" edges is of length $(m-2)$, but has negative degree $d(\theta) = -(m-2)$, as they manifestly extend in the direction opposite of the corresponding edges in the (convex) Newton polytope when $m \leqslant 2$.

**dim** $\theta, \theta^* = 2$: The faces $\theta^* = \theta^\triangledown$ in the spanning polytope $\Delta^\star_{F_3}$ all have nominal area and so unit degree, $d(\theta^*) = 1$. In order to asses $\theta \subset \Delta_{F_3}$ in the Newton polytope, we proceed in a manner similar to the analysis of the 2-faces for K3:

1. There are three large "vertical" and flip-folded (self-crossing) quadrangular faces $\theta \subset \Delta_{F_3}$ (analogous to $\Theta_2, \Theta_3 \subset \Delta_{\mathscr{F}_3}$ in Figure 3):

$$[(-1,-1,-1,-1),(-1,-1,-1,1+3m),(3,-1,-1,1-m),(3,-1,-1,-1)], \qquad (64a)$$
$$[(-1,-1,-1,-1),(-1,-1,-1,1+3m),(-1,3,-1,1-m),(-1,3,-1,-1)], \qquad (64b)$$
$$[(-1,-1,-1,-1),(-1,-1,-1,1+3m),(-1,-1,3,1-m),(-1,-1,3,-1)]. \qquad (64c)$$

The area of each of these three flip-folded (self-crossing) faces has two contributions in analogy with the similar calculation for the degree in the $n = 3$ case (see Figure 4), and is $d(\theta) = (2+3m)\cdot4+(2-m)\cdot4 = 8(2+m)$. The circumference is similarly given by $c(\theta) = 4+(2-m)+4+(2+3m)=2(6+m)$, since the degree of the vertical edge is $(2-m)$ and thus $\frac{1}{2}(d(\theta)-c(\theta)) = 2+3m$.

2. Next, we have three "horizontal" faces (analogous to $\Theta_4$ in Figure 3):

$$[(-1,-1,-1,-1),(3,-1,-1,-1),(-1,3,-1,-1)], \qquad (65a)$$
$$[(-1,-1,-1,-1),(3,-1,-1,-1),(-1,-1,3,-1)], \qquad (65b)$$
$$[(-1,-1,-1,-1),(-1,3,-1,-1),(-1,-1,3,-1)], \qquad (65c)$$

and three "slanted" faces (analogous to $\Theta_5$ in Figure 3):

$$[(-1,-1,-1,1+3m),(3,-1,-1,1-m),(-1,3,-1,1-m)], \qquad (66a)$$
$$[(-1,-1,-1,1+3m),(3,-1,-1,1-m),(-1,-1,3,1-m)], \qquad (66b)$$
$$[(-1,-1,-1,1+3m),(-1,3,-1,1-m),(-1,-1,3,1-m)]. \qquad (66c)$$

They are all simplices with the degree $d(\theta) = 4\cdot4 = 16$ since they have height four and base four. Since each of the edges in the above faces have degree 4, it immediately follows that $c(\theta) = 3\cdot4 = 12$, and so $\frac{1}{2}(d(\theta)-c(\theta)) = 2$.

3. Finally, the extension of the Newton polytope (analogous to $\Theta_1$ in Figure 3) is now a 3-sided prism. This has two triangular faces which contribute with positive degree:

$$[(3,-1,-1,-1),(-1,3,-1,-1),(-1,-1,3,-1)]_{\text{top}} \tag{67a}$$

$$\text{and } [(3,-1,-1,1-m),(-1,3,-1,1-m),(-1,-1,3,1-m)]_{\text{bottom}}, \tag{67b}$$

each having degree $d(\theta) = 16$ and circumference $c(\theta) = 12$, with $\frac{1}{2}(d(\theta) - c(\theta)) = 2$, as in the previous case. The three vertical rectangular walls of the 3-sided prism

$$[(3,-1,-1,-1),(-1,3,-1,-1),(-1,3,-1,1-m),(3,-1,-1,1-m)], \tag{68a}$$

$$[(3,-1,-1,-1),(-1,-1,3,-1),(-1,-1,3,1-m),(3,-1,-1,1-m)], \tag{68b}$$

$$[(-1,3,-1,-1),(-1,-1,3,-1),(-1,-1,3,1-m),(-1,3,-1,1-m)], \tag{68c}$$

have negative degrees, each equal to $d(\theta) = (2 \cdot (1-m) \cdot 4) = -8(m-2)$. They generalize the two vertical edges $[\mu_4, \mu_3]$ and $[\mu_6, \mu_5]$ in the $(n=3)$-dimensional case depicted in Figure 3. Similarly, the circumference is then given by $c(\theta) = 2 \cdot 4 + 2 \cdot (2-m) = 2(6-m)$. Thus, we have that $\frac{1}{2}(d(\theta) - c(\theta)) = 2 - 3m$

With the degrees for the 1- and 2-faces calculated, we first compute the Euler number. The contribution from the edges $\theta \subset \Delta_{F_3}$ and their polar 2-faces $\theta^* \subset \Delta^\star_{F_m}$ becomes

$$\sum_{\dim \theta = 1} d(\theta) d(\theta^*) = 1 \cdot (3m+2) + 3 \cdot (4) + 3 \cdot (4) + 2 \cdot 3 \cdot (4) + 3 \cdot [-(m-2)] = 56, \tag{69}$$

while that of the 2-faces $\theta \subset \Delta_{F_3}$ and their polar edge $\theta^* \subset \Delta^\star_{F_3}$ is similarly given by

$$\sum_{\dim \theta = 2} d(\theta) d(\theta^*) = 3 \cdot [8 \cdot (2+m)] + 3 \cdot 2 \cdot (16) + 2 \cdot (16) + 3 \cdot [-8 \cdot (m-2)] = 224. \tag{70}$$

Thus, we find that $\chi = 56 - 224 = -168$ in agreement with the gCICY result [1,5].

Next, we turn to the Hodge numbers, $h^{1,1}$ and $h^{2,1}$, respectively. For $n = 4$, there are $N_0^* = 6$ vertices in $\Delta^\star_{F_m}$ and $N_0 = 8$ vertices in $\Delta_{F_m}$. Thus, from our calculation of the degrees above we find

$$\sum_{\dim \theta = 1} d(\theta) + \sum_{\dim \theta^* = 1} d(\theta^*) = 1 \cdot (3m+2) + 12 \cdot 4 + 3 \cdot (2-m) + 14 \cdot 1 = 70; \tag{71}$$

$$N_0^* - N_1^* + \frac{1}{2} \sum_{\dim \theta = 2} \big( d(\theta) - c(\theta) \big) d(\theta^*) = 6 - 14 + \frac{1}{2} \big( (-2)56 \big) = -64; \tag{72}$$

$$N_0 - N_1 + \frac{1}{2} \sum_{\dim \theta = 2} \big( d(\theta) - c(\theta) \big) d(\theta^*) = 8 - 16 + 3(2+3m) + 6 \cdot 2 + 2 \cdot 2 + 3(2-3m) = 20. \tag{73}$$

Thus, it then follows from (40) and (39) that $h^{1,1} = 70 - 4 - 64 = 2$ and $h^{2,1} = 70 - 4 + 20 = 86$, again in agreement with the gCICY result [1,5].

As a further check, we can also calculate $h^{2,1}$ in the following way. Consider the Newton polytope for $F_m$, as discussed in Section 4.2. Excluding the $z = (1-m)$ and $z = -1$ triangles (where the extension intersects with positively oriented ordinary facets), this contains $15(m-3)$ integral points within the negatively oriented extension. In turn, it is straightforward to show by explicit counting that the positively oriented portion of the Newton polytope has $105 + \vartheta_3^m \, 15(m-3)$ integral points. These two $m$-dependent contributions therefore identically cancel for all $m$.

In turn, the corner edges of the extension,

$$\big\{ (3,-1,-1,z), \ (-1,3,-1,z), \ (-1,-1,3,z), \quad \text{with} \quad (1-m) \leqslant z \leqslant -1 \big\} \tag{74}$$

contain $3(m-1)$ points where the negatively oriented extension intersects with flip-folded (self-crossing) facets at the portions where those other facets are themselves negatively oriented. Owing to this double negative, these points count as deformations of the complex structure of $F_m$ itself, and are being canceled precisely by the surplus reparametrizations [5]. This renders the Hirzebruch 4-folds effectively rigid.

We thus remain with the effective number of 105 anticanonical sections, 18 reparametrizations and one overall scaling of the equation defining the hypersurface, producing the expected result: $105-18-1=86=h^{2,1}(Y_m)$ for the Calabi-Yau 3-fold $Y_m \subset F_m$ [5].

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
