# Peer review of "A Generalized Construction of Calabi-Yau Models and Mirror Symmetry"

_SciPost Physics, doi:SciPost Phys. 4, 009 (2018)_

## Round 2 · Referee Report · Anonymous · 2017-10-9

Strengths
1) novel construction of Calabi-Yau hypersurfaces in non-Fano toric varieties
2) clear presentation of this construction from GLSM point of view
3) proposal of combinatorial formula for Hodge numbers of such hypersurfaces
4) detailed exposition of an interesting class of examples which can also be presented as complete intersections in products of projective spaces
Weaknesses
1) not sufficiently self-contained, some crucial statements are not obvious from the presentation
2) some definitions and statements are not clear as stated
Report
This paper outlines a novel construction of Calabi-Yau manifolds as toric hypersurfaces by allowing the defining polynomial to be (specific type of) a meromorphic function of the homogeneous coordinates. This parallels earlier work on complete intersection Calabi-Yau manifolds for which a smooth compact Calabi-Yau can be obtained by allowing (some of) the defining polynomials to be meromorphic. Throughout this work, the authors treat a class of examples which can be constructed using both methods, i.e. which can be described as a complete intersection with meromorphic polynomials as well as a hypersurface in a toric variety defined by the vanishing locus of a meromorphic function. Hypersurfaces in toric varieties can be elegantly constructed using combinatorial objects called reflexive polyhedra. It is the main aim of the present work to provide a generalization of this construction to include cases for which the defining equation contains inverse powers of the homogeneous coordinates of the ambient toric variety. Besides the connection with complete intersections, the authors motivate their construction by describing it from the point of view of gauged linear sigma models. In particular, they start off by carefully examining the phase structure (enlarged Kaehler cone) of a model which has a geometric phase corresponding to an elliptic curve sitting in a Hirzebruch surface.
Requested changes
Although the paper is generally well-written and carefully guides the reader through the main class of examples, it has several shortcomings in presentation which I would like the authors to address prior to publication.
As mentioned above, the class of examples motivating the constructions of this paper have been previously discussed in the literature and were shown to give rise to smooth compact Calabi-Yau hypersurfaces.
1) To make the paper more self-contained, it would be appropriate if the authors could explain in some detail how the meromorphic defining polynomials define a smooth compact Calabi-Yau manifold. The statement "Thus f(x) defines a Calabi-Yau (n-1) hypersurface .." at the end of the first paragraph on p6 is not clear without further explanations, as the section f(x) in eq. 2b) has poles at x_{n+1}=0 and x_{n+2}=0 for m > 2.
Furthermore, I found the presentation of the construction in terms of VEX polytope slightly confusing in several places:
2) In the discussion at the beginning of section 3, the \nu_\rho are not all vertices of Delta^*, but only some of them are.
3) The polar \Delta^\circ defined in eq 15 is in general not a lattice polytope, in which case it cannot be spanning polytope. If it is a convex polytope, however, \Delta and \Delta^\circ must be reflexive pair.
4) In construction 3.1. d(\theta) is not defined at this point in the paper. I believe the definition is found in Appendix B, p21. Furthermore, I do not see how the star triangulation of \Delta is relevant at this point as it is not used in the following points 1 to 3.
5) Construction 3.1. point 1: how is such a decomposition found ? Is it unique ?
6) Construction 3.1. point 2: The duality relation for faces is not good enough to fix a dual face, but must be combined with a further relation. For reflexive pairs
\langle \Delta, \Delta^\circ \rangle \geq -1 (1)
together with
\langle face on Delta, \face on \Delta^\circ \rangle = -1 (2)
defines a pair of dual faces. What is the analogue of relation (1) in the present case ? Once the dual pair of non-convex polytopes is found, (2) together with the fact that the dual pair are faces on the pair of polytopes is good enough, but here it is used to define the dual polytope.
7) Construction 3.1. point 3: Why can we assemble these faces into a (non-convex) polyhedron ?
8) A suggestion: in the construction of VEX polytopes, convexity is dropped. The requirement of existence of a star triangulation seems to be more efficiently expressed as saying that such non-convex polytopes \Delta still define a star-domain in R^n, i.e. connecting every point on \Delta to the origin gives a line contained in \Delta.

---

## Round 2 · Referee Report · Anonymous · 2017-10-17

Strengths
This is a novel and well written paper exploring new constructions of Calabi-Yau manifolds. The proposed generalized construction of toric hypersurfaces (in so-called VEX polytopes) is innovative and potentially a very valuable new contribution to the study of string compactifications.
The authors do a very good job of introducing a complex subject in detail with many practical examples presented.
Overall, I would certainly recommend this paper for publication.
Weaknesses
1. Unclear discussion of smoothness/transversality differentiability
2. The above point is reflected in open questions about the vacuum space (and possible non-compact directions) in the (0,2) GLSMs (including the possibilities of extra light states, UV completions, etc).
Report
This is a very nice, creative and potentially broadly important paper and I would recommend it for publication.
However, there is one aspect of the work that I would like to see clarified a little before publication and this regards singularities of the CY hypersurface. In general, the authors employ Laurent monomials in the defining equations of the toric hypersurfaces (equivalently in the superpotentials of generalized (0,2) GLSMs).
They comment on the "transversality" of these hypersurfaces in several places throughout the work but it is not at all clear that the definition (f=0 and df=0 share no common locus) is at all the correct notion for the construction at hand. In particular, no analysis shows that the derivative "df" is even well defined for the Laurent polynomials given (i.e. the functions may not be differentiable everywhere in parameter/field space), much less that the associated toric hypersurfaces are smooth. On this front it is not at all clear that the analysis of Appendix A is the correct one to consider or useful to argue smoothness.
This same issue affects the analysis of the (0,2) GLSM vacuum space -- i.e. the presence of extra light states, non-compact branches to the vacuum space or possible UV completions.
It may be that a full analysis of which hypersurfaces in VEX polytopes are smooth is beyond the scope of the current paper (and I would still recommend this work for publication without it). However, it would be good to see these important open questions more clearly flagged in the present work.
Requested changes
1. Clearly address the differentiability of the Laurent polynomials given
2. Clarify whether or not transversality is a relevant/sufficient notion.
3. Flag importance of smoothness for new construction of CY manifolds (and perhaps summarize how this might be addressed in future).

---

## Round 3 · Referee Report · Anonymous · 2017-11-26

Strengths

The strengths remain as stated in my initial report

Weaknesses

The weaknesses in the paper have been largely addressed by the authors in the revised version.

Report

This revision meets my expectations and I recommend the work for publication.

---

## Round 3 · Referee Report · Anonymous · 2017-11-30

Strengths

As in my previous report

Weaknesses

none

Report

The authors have carefully revised their paper and significantly clarified the discussion. In particular, all of the shortcomings noted in my previous report have been well addressed.

---

## Round 3 · Author Response

We have revised the article throughout, responding to all of the Referees' comments and questions (see below), and have used the opportunity to further clarify the narrative and update the bibliography.

---

## Round 3 · List of Changes

To the Referee 1:

1) We have thoroughly rewritten Appendix A, expanding to spell out the motivations and details of the "working definition" that has been used throughout the analysis reported in this article. We also clarify (in several places including the conclusions) that we have verified, wherever possible, that the so-obtained results are both self-consistent, and also consistent with the by now well established results of Refs. [1,5,6].

2) In the text preceding Eq.(16), we have changed the specification of $\nu_\rho$ from "...vertices..." to "...the $N$-integral points $\nu_\rho$ of which being the minimal generators..." as this covers also the exceptional $m=2$ case, when $\nu_1=(-1,\dots,-1,0)$ is in a face, corresponding to the MPCP exceptional set.

3) The sentence containing Eq.(15) now correctly qualifies its use for reflexive (and convex) polytopes.

4) The definition is inserted in footnote 9, at the first mention of the "degree," on p.9. Its effective use in counting is mentioned right after Construction 3.1.

5) We have rewritten Construction 3.1 so as to refer to the full disjoint union of all faces, with the non-convex ones subdivided into convex ones. In our experience, this is very redundant, but does not introduce undue quandaries of non-uniqueness and their relevance. In turn, the subdivision of non-convex k-faces into convex k-faces poses no problems, since they are all k-coplanar so that their polars coincide. This has been clarified in the revised document.

6) The trans-polar polytope is assembled "from ground up," using the particular "dually" implied relations, which are now spelled out as Eqs. (17) at the end of Step 3 in Construction 3.1. Their use is illustrated in the explicit construction in Section 3.2, and we amplified the discussion of the observed sufficiency of the specifications in Construction 3.1 in the narrative on p.9-10.

7) As now clarified in several places, we do not have a proof that the specifications of Construction 3.1 always suffice, but have observed that they do in dozens upon dozens of examples (including some infinite sequences) purposefully invented to test it. Indeed, we wholeheartedly welcome any and all attempt to either prove or improve Construction 3.1 --- and in a twin fashion also provide a conclusive definition of VEX polytopes as a maximal closure of trans-polar pairs of polytopes.

8) We have included an explicit reference to star-domains, noting however that they must be understood in a generalized fashion, implicit in the (cited) literature on "multi-fans."

To Referee 2:

1. We have thoroughly rewritten Appendix A, expanding to spell out the motivations and details of the "working definition" that has been used throughout the analysis reported in this article. We also clarify (in several places including the conclusions) that we have verified, wherever possible, that the so-obtained results are both self-consistent, and also consistent with the by now well established results of Refs. [1,5,6].

2. In the rewritten Appendix A (as well as in Section 2), we highlight the places where and why the related notions of "base-locus" and "transversality" (and Batyrev's $\Delta$-regularity) turn up. We now also explicitly state that our article addresses the low-energy regime of the considered GLSMs, and defer the UV analysis.

3. The related issues of "intrinsic limits," transversality and non-singularity are detailed in Appendix A, and noted also elsewhere throughout this revision.

---

## Editorial Decision

published